



# A Meridionally Averaged Model of Eastern Boundary Upwelling Systems (MAMEBUSv1.0)

Jordyn E. Moscoso[1], Andrew L. Stewart[1], Daniele Bianchi[1], and James C. McWilliams[1]

[1]Department of Atmospheric Sciences – University of California, Los Angeles

**Correspondence:** Jordyn Moscoso (jmoscoso@atmos.ucla.edu)

**Abstract.** Eastern Boundary Upwelling Systems (EBUSs) are physically and biologically active regions of the ocean with substantial impacts on ocean biogeochemistry, ecology, and global fish catch. Previous studies have used models of varying complexity to study EBUS dynamics, ranging from minimal two-dimensional (2D) models to comprehensive regional and global models. An advantage of 2D models is that they are more computationally efficient and easier to interpret than compre-
hensive regional models, but their key drawback is the lack of explicit representations of important three-dimensional processes that control biology in upwelling systems. These processes include eddy quenching of nutrients and meridional transport of nutrients and heat. The authors present a Meridionally Averaged Model of Eastern Boundary Upwelling Systems (MAMEBUS) that aims at combining the benefits of 2D and 3D approaches to modeling EBUSs by parameterizing the key 3D processes in a 2D framework. MAMEBUS couples the primitive equations for the physical state of the ocean with a nutrient-phytoplankton-
zooplankton-detritus model of the ecosystem, solved in terrain following coordinates. This article defines the equations that describe the tracer, momentum, and biological evolution, along with physical parameterizations of eddy advection, isopycal mixing, and boundary layer mixing. It describes the details of the numerical schemes and their implementation in the model code, and provides a reference solution validated against observations from the California Current. The goal of MAMEBUS is to facilitate future studies to efficiently explore the wide space of physical and biogeochemical parameters that control the
zonal variations in EBUSs.

## 1 Introduction

Eastern Boundary Upwelling Systems (EBUSs) are among of the most biologically productive regions in the ocean, supporting diverse ecosystems, and contributing to a significant portion of the global fish catch (Bakun and Parrish, 1982). The character-istic wind-driven upwelling dominant in EBUSs is forced by an equatorward meridional wind stress that decreases toward the
shore, driving a zonal Ekman transport offshore. The resulting Ekman pumping brings cold, nutrient-rich water to the surface, fueling primary productivity (Jacox and Edwards, 2012; Chavez and Messié, 2009; Rykaczewski and Dunne, 2010).

The upwelling-favorable winds also drive baroclinic, equatorward geostrophic current, which sheds mesoscale eddies (Colas et al., 2013). Together with offshore Ekman transport, mesoscale eddies redistribute nutrients zonally and subduct nutrients and other tracers into the ocean subsurface (Capet et al., 2008; Gruber et al., 2011; Renault et al., 2016). The resulting cross-shore
gradient of nutrients at the surface supports a zonal variation in the abundance of phytoplankton, with high biomass and





chlorophyll nearshore, and low offshore (Chavez and Messié, 2009). The size structure of phytoplankton is similarly affected, with larger cells with higher nutrient demand onshore, and smaller cells offshore (Cabre et al., 2013).

While these qualitative patterns of productivity are common to upwelling systems, previous studies have shown that productivity varies substantially between EBUSs, but the causes of these inter-EBUS variations are not well understood. Possible

physical drivers of these inter-EBUS variations include the shape and strength of the wind stress curl, which set the upwelling strength and source depth (Bakun and Nelson, 1991; Jacox and Edwards, 2012). This in turn controls the energy transferred to the baroclinic eddy field, modulating surface nutrient availability via the "eddy quenching" mechanism (Gruber et al., 2011; Renault et al., 2016). Additionally, inter-EBUS variations may have biogeochemical origins, for example due differing subsurface oxygen inventories (Chavez and Messié, 2009).

Our understanding of these drivers is hindered in part by the observational limitations, and in part by the computational expense of regional models that can resolve the processes mentioned above. A range of models of varying complexity have been used to study EBUSs, from minimal two-dimensional (2D) models (Jacox and Edwards, 2012; Jacox et al., 2014) to comprehensive regional models (Shchepetkin and McWilliams, 2005; Chenillat et al., 2018). While 2D models require fewer computational resources than comprehensive regional model studies, and thus allow a more comprehensive exploration of the

relevant parameter space, they lack the explicit representation of important physical processes that affect biology in upwelling systems (i.e. eddy-quenching and meridional transport of nutrients).

Here, we aim to close the current gap in understanding by developing an idealized, quasi-2D model of the physics and biogeochemistry of EBUSs. The model includes parameterizations of the key three-dimensional processes, while retaining the computational efficiency of a 2D model. The model is cast in a residual-mean framework (Plumb and Ferrari, 2005a) in terrain

following coordinates (Song and Haidvogel, 1994), and is referred to as the Meridionally Averaged Model of Eastern Boundary Upwelling Systems (MAMEBUS). A schematic of all the important processes in MAMEBUS is shown in Figure 1.

The rest of the paper is organized as follows. In Section 2, we describe the equations and physical parameterizations implemented in MAMEBUS, including general formulation of tracer advection and diffusion, the time-dependent turbulent thermal wind approximation of the momentum equations (T3W), eddy and boundary layer parameterizations, and our ecosystem

formulation. In Section 3, we detail the algorithms and discretizations, including mesh specification, vertical coordinate transformation, and time integration. In Section 4, we describe the implementation of MAMEBUS including the various options available to the user, parameter choices, initialization, and output. In Section 5, we describe reference solutions for MAMEBUS, discussing model sensitivities to changes in bathymetry, wind forcing, and surface heat fluxes. Finally, in Section 6 we discuss further model development and future work.

## 2    MAMEBUS Framework

MAMEBUS is comprised of a series of components that are necessary to capture physical-biogeochemical dynamics in EBUSs:
(1) explicit momentum conservation in form of geostrophic, hydrostatic, and Ekman balances implemented as part of the T3W

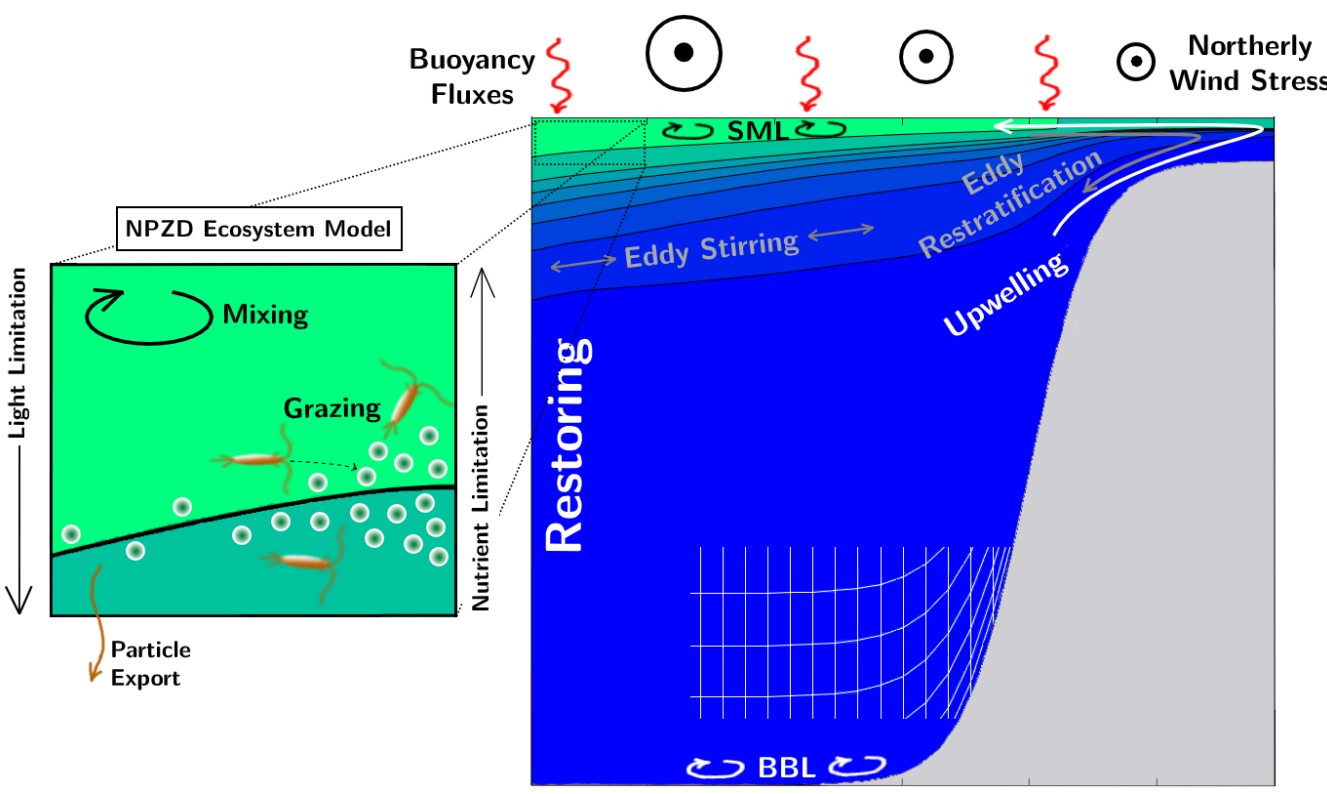

**Figure 1.** A schematic of the essential components of the Meridionally Averaged Model of Eastern Bounday Upwelling Systems (MAME-BUS). This schematic highlights some components that the user is able to control including the offshore restoring conditions, the eddy mixing along isopycnals, the wind forcing, the surface mixed layer and bottom boundary layer parameterizations and grid spacing.

formulation; (2) eddies and their effect on material transport; (3) surface and bottom boundary layers; (4) nutrient and plankton cycles in form of a size-structured "NPZD"-type model (Banas, 2011).

   With the exception of the velocity field, all tracers in MAMEBUS evolve according to the following conservation equation:

$$\frac{\partial \bar{c}}{\partial t} = \frac{\partial \bar{c}}{\partial t}\bigg|_{\text{phys}} + \frac{\partial \bar{c}}{\partial t}\bigg|_{\text{bio}} + \frac{\partial \bar{c}}{\partial t}\bigg|_{\text{nct}}, \tag{1}$$

5  where the bar indicates a meridional average. The key physical tracer that follows Equation (1) is temperature, $\theta$, which serves as the thermodynamic variable in our model. We choose temperature as our thermodynamic variable because of its important





effects on biogeochemistry (Sarmiento and Gruber, 2006). The biogeochemical tracers that are affected by the biogeochemical evolution term, $\partial_t \bar{c}\big|_{\text{bio}}$, are a limiting nutrient N (here expressed in nitrogen units, akin to nitrate); a phytoplankton tracer, P; a zooplankton tracer, Z; and a detrital pool, D. The non-conservative terms, $\partial_t \bar{c}\big|_{\text{nct}}$, represent physical sources and sinks of tracers, including surface fluxes, restoring at the offshore boundary, and optional restoring throughout the domain.

## 2.1 Tracer evolution

We first formulate an evolution equation for the meridionally-averaged concentration of an arbitrary tracer $c$. We assume that $c$ evolves according to a combination of advection by the three-dimensional ocean flow and diffusion by microscale mixing processes,

$$\frac{\partial c}{\partial t}\bigg|_{\text{phys}} = \underbrace{-\nabla_3 \cdot (\mathbf{u}_3 c)}_{\text{advection}} + \underbrace{\nabla_3 \cdot (\kappa_{\text{dia}} \nabla_3 c)}_{\text{mixing}}, \tag{2}$$

Here $\mathbf{u}_3$ is the three-dimensional velocity vector, $\nabla_3$ is the three-dimensional gradient operator, and $\kappa_{\text{dia}}$ the microscale diffusivity. In (2) we have assumed that the velocity field is nondivergent, *i.e.* $\nabla_3 \cdot \mathbf{u}_3 = 0$. We further assume that $\mathbf{u}_3$ and $c$ have already been averaged over a short timescale to exclude fluctuations associated with microscale eddies, whose effects are parameterized via the microscale mixing term (*e.g.* Aiki and Richards, 2008). We further simplifiy (2) by assuming that horizontal tracer gradients are small compared with vertical gradients, *i.e.* $\partial_z c \gg \partial_x c, \partial_y c$, as is typical for oceanic scales of evolution (*e.g.* Vallis, 2017). This implies that the microscale mixing acts primarily in the vertical, *i.e.*,

$$\frac{\partial c}{\partial t}\bigg|_{\text{phys}} \approx -\nabla_3 \cdot (\mathbf{u}_3 c) + \frac{\partial}{\partial z}\left(\kappa_{\text{dia}} \frac{\partial c}{\partial z}\right). \tag{3}$$

We now reduce the dimensionality of (3) by taking a meridional average, which we denote via an overbar,

$$\overline{\bullet} = \frac{1}{L_y} \int\limits_0^{L_y} \bullet \, \mathrm{d}y. \tag{4}$$

Here $L_y$ is the meridional length of the region of interest and $y$ is the meridional coordinate. Though we refer to this average as "meridional" throughout the text, for the purpose of comparison with EBUSs in nature this average might be thought of instead as an along-coast average, or as an average following isobaths, under the assumption that the additional metric terms introduced by such coordinate transformations are negligible. We next perform a Reynolds decomposition of the velocity and tracer fields,

$$\mathbf{u} = \overline{\mathbf{u}} + \mathbf{u}', \tag{5a}$$

$$c = \overline{c} + c', \tag{5b}$$

where primes $'$ denote perturbations from the meridional average. Taking a meridional average of (3) then yields

$$\frac{\partial \overline{c}}{\partial t}\bigg|_{\text{phys}} = \underbrace{-\nabla \cdot (\overline{\mathbf{u}}\,\overline{c})}_{\text{mean advection}} \underbrace{-\nabla \cdot (\overline{\mathbf{u}'c'})}_{\text{eddy flux}} \underbrace{-\frac{1}{L_y}\left[vc\right]_0^{L_y}}_{\text{meridional advection}} + \underbrace{\frac{\partial}{\partial z}\left(\kappa_{\text{dia}} \frac{\partial c}{\partial z}\right)}_{\text{mixing}}. \tag{6}$$



Here we have used (5a)–(5b) and the property that perturbations vanish under the average, *i.e.* $\overline{\mathbf{u}'} = \overline{c'} = 0$. We further define $\nabla \equiv \partial_x \hat{\mathbf{x}} + \partial_z \hat{\mathbf{z}}$ as the zonal/vertical gradient operator, and $\mathbf{u} = u\hat{\mathbf{x}} + w\hat{\mathbf{z}}$ as the zonal/vertical velocity vector. The square bracket indicates the difference between $vc$ at the northern and southern boundaries of the domain of integration, *i.e.*

$$\left[vc\right]_0^{L_y} = vc\Big|_{y=L_y} - vc\Big|_{y=0}. \tag{7}$$

In its current form, Equation (6) cannot be solved prognostically for $\bar{c}$ because it includes correlations between perturbation quantities, *i.e.* the eddy tracer flux $\overline{\mathbf{u}'c'}$. Assuming that these perturbations are associated with mesoscale eddies, we parameterize the eddy tracer flux following Gent and McWilliams (1990) and Redi (1982). Specifically, we decompose the eddy tracer flux into advection of the mean tracer $\bar{c}$ by "eddy-induced velocity" $\mathbf{u}^\star$ and diffusion of $\bar{c}$ along the mean buoyancy surfaces (see Burke et al., 2015),

$$\nabla \cdot \left(\overline{\mathbf{u}'c'}\right) = \nabla \cdot \left(\mathbf{u}^\star \bar{c}\right) - \nabla \cdot \left(\kappa_{\text{iso}} \nabla_\parallel \bar{c}\right). \tag{8}$$

Here $\nabla_\parallel$ denotes the gradient along mean buoyancy surfaces (see §2.2. A more detailed derivation of (8) is given in Appendix A. We additionally simplify the meridional tracer advection term by assuming that $\overline{\partial v / \partial y} \approx 0$, *i.e.* that the meridional tracer flux convergence is dominated by meridional tracer gradients, and that correlations between $\kappa_{\text{dia}}$ and $c$ are negligible, *i.e.* that the meridionally averaged vertical diffusive tracer flux serves to diffuse $\bar{c}$ downgradient. With these simplifications, the

full equation for the physical evolution of tracers is given by,

$$\frac{\partial \bar{c}}{\partial t}\bigg|_{\text{phys}} = \underbrace{-\nabla \cdot (\overline{\mathbf{u}}\bar{c})}_{\text{mean advection}} \underbrace{-\frac{\overline{v}}{L_y}\left[c\right]_0^{L_y}}_{\text{meridional advection}} \underbrace{-\nabla \cdot (\mathbf{u}^*\bar{c})}_{\text{eddy advection}} \underbrace{-\nabla \cdot \left(\kappa_{\text{iso}} \nabla_\parallel \bar{c}\right)}_{\text{eddy stirring}} + \underbrace{\partial_z \left(\kappa_{\text{dia}} \partial_z \bar{c}\right)}_{\text{mixing}}. \tag{9}$$

The terms on the right-hand side of (9) are discussed further in the following sections: in Section 2.2 we discuss the evolution of the mean velocity $\overline{\mathbf{u}}$ via the momentum equations, and in Section 2.3 we discuss the sub-gridscale parameterizations, *i.e.* eddy advection, eddy stirring and mixing.

## 2.2 Momentum Evolution Equations

To evolve a meridionally-averaged tracer $\bar{c}$ using (9), the meridionally-averaged velocity field $\overline{\mathbf{u}}_3$ is required. This velocity field is evolved in MAMEBUS by solving a simplified form of the hydrostatic Boussinesq momentum and continuity equations with a linear equation of state (Vallis, 2017),

$$\frac{\partial u}{\partial t} = -\mathbf{u}_3 \cdot \nabla_3 u + fv - \frac{\partial \phi}{\partial x} + \frac{\partial}{\partial z}\left(\kappa_{\text{dia}} \frac{\partial u}{\partial z}\right), \tag{10a}$$

$$\frac{\partial v}{\partial t} = -\mathbf{u}_3 \cdot \nabla_3 v - fu - \frac{\partial \phi}{\partial y} + \frac{\partial}{\partial z}\left(\kappa_{\text{dia}} \frac{\partial v}{\partial z}\right), \tag{10b}$$

$$\frac{\partial \phi}{\partial z} = b, \tag{10c}$$

$$\nabla_3 \cdot \mathbf{u}_3 = 0, \tag{10d}$$

$$b = g\alpha\theta. \tag{10e}$$


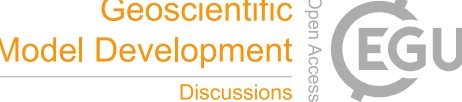

Here, $\phi = p/\rho_0$ is the dynamic pressure, where $\rho_0$ is an arbitrary reference density, $b$ is the buoyancy, $\theta$ is the potential temperature, $\alpha$ is the thermal expansion coefficient (assumed constant), $g$ is the gravitational constant and $f$ is the Coriolis parameter. Note that we have assumed that momentum is mixed by microscale turbulence following the same diffusivity $\kappa_{\mathrm{dia}}$ as tracers (see Section 2.1), *i.e.* that the turbulent Prandtl number (*e.g.* Kays, 1994) is exactly equal to one.

As in Section 2.1, we now meridionally average (10a)–(10e) to obtain evolution equations for $\overline{u}$ and $\overline{v}$, and thus implicitly also for $\overline{w}$. This yields the following set of averaged equations:

$$\frac{\partial \overline{u}}{\partial t} = f\overline{v} - \frac{\partial \overline{\phi}}{\partial x} + \frac{\partial}{\partial z}\left(\kappa_{\mathrm{dia}}\frac{\partial \overline{u}}{\partial z}\right), \tag{11a}$$

$$\frac{\partial \overline{v}}{\partial t} = -f\overline{u} - \frac{1}{L_y}\left[\phi\right]_0^{L_y} + \frac{\partial}{\partial z}\left(\kappa_{\mathrm{dia}}\frac{\partial \overline{v}}{\partial z}\right), \tag{11b}$$

$$\nabla \cdot \overline{\mathbf{u}} = 0, \tag{11c}$$

$$\frac{\partial \overline{\phi}}{\partial z} = \overline{b}, \tag{11d}$$

$$\overline{b} = g\alpha\overline{\theta}. \tag{11e}$$

Here, we have made the frictional-geostrophic approximation (*e.g.* Edwards et al., 1998), assuming that the Rossby number of the flow is small (*e.g.* Vallis, 2017) and thus that momentum advection (second terms from the left in (10a)–(10b)) is negligible compared to other terms in the momentum equation. However, we have retained the time-evolution terms (leftmost terms in

(10a)–(10b)) to allow forward evolution of the horizontal velocity fields; if these terms were neglected then these terms would need to be computed diagnostically at each time step. The resulting system is almost identical to the time-dependent turbulent thermal wind (T3W) equations (Dauhajre and McWilliams, 2018), a time-varying extension of the turbulent thermal wind balance (Gula et al., 2014), which was developed to explain the circulation of submesoscale fronts. The meridional pressure gradient in (11a) is imposed, rather than solved for prognostically, and is assumed to be set by the larger-scale subtropical gyre

circulation encompassing the EBUS, which expicitly differs from the work done in Dauhajre and McWilliams (2018) which focuses on more rapid time varying evolution on smaller scales. Together with the tracer advection equation for potential temperature (*i.e.* (9) with $c = \theta$), (11a)–(11e) comprise a closed set of equations for the physical evolution of MAMEBUS.

In (11c) we have invoked the earlier assumption that $\overline{\partial v/\partial y} \approx 0$ (see Section 2.1), such that the averaged velocity field is nondivergent in the $x/z$ plane. This implies that the zonal/vertical velocity field can be related to a mean streamfunction $\overline{\psi}$ via

$$\overline{u} = -\frac{\partial \overline{\psi}}{\partial z}, \qquad \overline{w} = \frac{\partial \overline{\psi}}{\partial x}. \tag{12}$$

These relationships allow us to calculate $\overline{\psi}$, and thus $\overline{w}$, from $\overline{u}$, subject to the boundary conditions

$$\overline{\psi} = 0 \quad \text{at} \quad z = 0, z = \overline{\eta}_b(x). \tag{13}$$

Here $z = \overline{\eta}_b(x)$ is the mean sea floor elevation.

Additional boundary conditions are required to solve (11a)–(11e) prognostically. Specifically, we require that the vertical

turbulent stress in (11a)–(11b) matches the wind stress applied at the sea surface and the drag stress at the sea floor, with the





latter formulated via a linear drag law. Formally, these boundary conditions are

$$\kappa_{\mathrm{dia}}\frac{\partial \overline{u}}{\partial z} = 0, \qquad \kappa_{\mathrm{dia}}\frac{\partial \overline{v}}{\partial z} = \frac{\overline{\tau}^y}{\rho_0} \qquad \text{at } z = 0, \tag{14a}$$

$$\kappa_{\mathrm{dia}}\frac{\partial \overline{u}}{\partial z} = r\overline{u}, \qquad \kappa_{\mathrm{dia}}\frac{\partial \overline{v}}{\partial z} = r\overline{v} \qquad \text{at } z = \overline{\eta}_b(x). \tag{14b}$$

Here $r$ is a linear drag coefficient and $\overline{\tau}^y$ is the meridional wind-stress.

## 2.3 Physical parameterizations

In this section we describe the parameterization of unresolved microscale mixing in the tracer evolution Equation (9) and the horizontal momentum Equations (11a)–(11b), and of mesoscale eddy advection and stirring in (9). This amounts to parameterizing the diapycnal diffusivity $\kappa_{\mathrm{dia}}$, the isopycnal diffusivity $\kappa_{\mathrm{iso}}$, and the eddy velocity $\mathbf{u}^\star$.

### 2.3.1 Diapycnal Mixing

We formulate the diapycnal mixing coefficient $\kappa_{\mathrm{dia}}$ as a sum of four distinct contributing processes: surface mixed layer turbulence ($\kappa_{\mathrm{sml}}$), bottom boundary layer turbulence ($\kappa_{\mathrm{bbl}}$), turbulence due to convective overturns within the water column ($\kappa_{\mathrm{conv}}$), and background mixing due to internal wave breaking ($\kappa_{\mathrm{bg}}$). Formally, we write

$$\kappa_{\mathrm{dia}}(x,z,t) = \kappa_{\mathrm{sml}}(x,z) + \kappa_{\mathrm{bbl}}(x,z) + \kappa_{\mathrm{conv}}(x,z,t) + \kappa_{\mathrm{bg}}(x,z). \tag{15}$$

The terms on the right-hand side of (15) are discussed in turn in the following paragraphs.

The diapycnal diffusivity in the surface mixed layer, $\kappa_{\mathrm{sml}}$, is prescribed to have the same structure as that used in the K-profile parameterization (KPP) of Large et al. (1994). However, for simplicity, the mixed layer depth $H_{\mathrm{sml}}(x)$ and maximum magnitude $\kappa_{\mathrm{sml}}(x)$ are prescribed functions, rather than depending on the local surface forcing. The vertical profile of $\kappa_{\mathrm{dia}}$ in the surface mixed layer, *i.e.* $-H_{\mathrm{sml}} < z < 0$, is given by

$$\kappa_{\mathrm{sml}}(x,z) = \kappa_{\mathrm{sml}}^0 G_{\mathrm{KPP}}(\sigma_{\mathrm{sml}}), \tag{16}$$

where the dimensionless surface mixed layer vertical coordinate $\sigma_{\mathrm{sml}} = -z/H_{\mathrm{sml}}$ is defined such that $0 \le \sigma_{\mathrm{sml}} \le 1$ within the mixed layer. The structure function $G_{\mathrm{KPP}}(\sigma_{\mathrm{sml}})$ is given by,

$$G_{\mathrm{KPP}}(\sigma) = \begin{cases} \dfrac{27}{4}\sigma_{\mathrm{sml}}(1-\sigma_{\mathrm{sml}})^2, & 0 \le \sigma_{\mathrm{sml}} \le 1, \\[2mm] 0, & \sigma_{\mathrm{sml}} \ge 1, \end{cases} \tag{17}$$

following Large et al. (1994) and Troen and Mahrt (1986). The scaling factor $27/4$ ensures that $G_{\mathrm{KPP}}(\sigma_{\mathrm{sml}})$ has a maximum of 1 for $0 < \sigma_{\mathrm{sml}} < 1$.

The diapycnal diffusivity in the bottom boundary layer, $\kappa_{\mathrm{bbl}}$, is prescibed in the same way as $\kappa_{\mathrm{sml}}$, but over the depth range $\overline{\eta}_b < z < \overline{\eta}_b + H_{\mathrm{bbl}}(x)$. Thus, analogous to (16), we prescribe

$$\kappa_{\mathrm{bbl}}(x,z) = \kappa_{\mathrm{bbl}}^0 G_{\mathrm{KPP}}(\sigma_{\mathrm{bbl}}), \tag{18}$$





where the dimensionless bottom boundary layer vertical coordinate is defined as $\sigma_{\mathrm{bbl}} = (z - \overline{\eta}_b)/H_{\mathrm{bbl}}$.

At any point in space and time at which the water column is statically unstable, *i.e.* when $N^2 < 0$, we increase the value of $\kappa_{\mathrm{dia}}$ is increased locally to parameterize the effect of density-driven convection. That is, we prescribe $\kappa_{\mathrm{conv}}$ following

$$\kappa_{\mathrm{conv}} = \begin{cases} \kappa_{\mathrm{conv}}^0, & N^2 < 0, \\ 0, & N^2 \geq 0. \end{cases} \tag{19}$$

Finally, the background diapycnal mixing, $\kappa_{\mathrm{bg}}(x, z)$, is simply prescribed as a constant background diffusivity. There are other that can be used (e.g. St. Laurent et al. (2002)), but we opt for simplicity in the first version of this model.

### 2.3.2 Eddy advection and isopycnal mixing

We now discuss the formulation of the eddy advection and isopycnal mixing terms in (9). As discussed in Section 2.1, we follow the assumptions and formalism of the Gent and McWilliams (1990) and Redi (1982) parameterizations, which are commonly
used in ocean models that do not explicitly resolve mesoscale eddies (*e.g.* Gent, 2011). These parameterizations assume that eddy-induced fluxes of buoyancy and tracer diffusion are directed along isopycnal slopes, and so must be augmented in the ocean's surface mixed layer (SML) and bottom boundary layer (BBL). Here the isopycnal slopes become very steep and isopycnals incrop at the sea surface and floor (Tréguier et al., 1997). MAMEBUS therefore uses a modified form of the Ferrari et al. (2008) boundary layer parameterization, in which eddy buoyancy and tracer fluxes are rotated through the SML and
BBL in order to enforce vanishing eddy-induced mass and tracer fluxes through the boundaries. Here we summarize salient properties of this scheme, and in Appendix C we highlight differences between our scheme and that of Ferrari et al. (2008).

The eddy-induced velocity $\mathbf{u}^\star = (u^\star, w^\star)$, introduced in (8), is nondivergent by construction (see Appendix A) and so we write it as

$$u^\star = -\frac{\partial \psi^\star}{\partial z}, \qquad w^\star = \frac{\partial \psi^\star}{\partial x}, \tag{20}$$

where $\psi^\star$ is the "eddy streamfunction". This advecting streamfunction is assumed to be the same for all tracers, which is accurate in the limit of small-amplitude fluctuations of the velocity and tracer fields (Plumb, 1979), and takes the form

$$\psi^\star = \kappa_{\mathrm{gm}} S_{\mathrm{gm}}. \tag{21}$$

Here $\kappa_{\mathrm{gm}}$ is the Gent–McWilliams diffusivity and the $S_{\mathrm{gm}}$ is the is the Gent–McWilliams slope. The latter is conventionally set equal to the mean isopycnal slope (Gent and McWilliams, 1990),

$$S_{\mathrm{int}} = -\partial_x \overline{b}/\partial_z \overline{b}. \tag{22}$$

However we allow $S_{\mathrm{gm}}$ to diverge from $S_{\mathrm{int}}$ in the SML and BBL, in part to ensure that the no-flux surface and bottom boundary conditions are satisfied (Ferrari et al., 2008)

$$\psi^\star = 0 \qquad \text{at} \qquad z = 0, \ z = \overline{\eta}_b(x). \tag{23}$$





Specifically, we prescribe

$$
S_{\mathrm{gm}} =
\begin{cases}
S_{\mathrm{sml}}, & -H_{\mathrm{sml}}(x) < z < 0, \\
S_{\mathrm{int}}, & \overline{\eta}_b(x) + H_{\mathrm{bbl}}(x) < z < -H_{\mathrm{sml}}(x), \\
S_{\mathrm{bbl}}, & \overline{\eta}_b(x) < z < \overline{\eta}_b(x) + H_{\mathrm{bbl}}(x),
\end{cases}
\tag{24}
$$

The formulation of the modified slopes $S_{\mathrm{sml}}$ and $S_{\mathrm{bbl}}$ are discussed below in Sections 2.3.2.1 and 2.3.2.2.

The isopycnal mixing operator serves to mix tracers down their mean gradients, in a direction that is parallel to mean

isopycnal surfaces in the ocean interior, following Redi (1982). This may be written component-wise as

$$
\nabla \cdot \left( \kappa_{\mathrm{iso}} \nabla_{\parallel} \overline{c} \right) = \frac{\partial}{\partial x} \left( \kappa_{\mathrm{iso}} \frac{\partial \overline{c}}{\partial x} + \kappa_{\mathrm{iso}} S_{\mathrm{iso}} \frac{\partial \overline{c}}{\partial z} \right) + \frac{\partial}{\partial z} \left( \kappa_{\mathrm{iso}} S_{\mathrm{iso}} \frac{\partial \overline{c}}{\partial x} + \kappa_{\mathrm{iso}} S_{\mathrm{iso}}^2 \frac{\partial \overline{c}}{\partial z} \right),
\tag{25}
$$

where $S_{\mathrm{iso}}$ denotes the slope of the surface along which the tracer is to be mixed and is assumed to be small ($S_{\mathrm{iso}} \ll 1$). Similar to $S_{\mathrm{gm}}$, this slope is conventionally set equal to the mean isopycnal slope $S_{\mathrm{int}}$, but we apply modifications to the formulation of $S_{\mathrm{iso}}$ in the SML and BBL to ensure that there is zero eddy-induced tracer flux through the domain boundaries, *i.e.*

$\kappa_{\mathrm{iso}} \nabla_{\parallel} \overline{c} \cdot \hat{\mathbf{n}} = 0 \qquad \text{at} \qquad z = 0,\, z = \overline{\eta}_b(x),$
$\tag{26}$

where $\hat{\mathbf{n}}$ is a unit vector oriented perpendicular to the sea surface or sea floor. Specifically, we prescribe

$$
S_{\mathrm{iso}} =
\begin{cases}
S_{\mathrm{sml}}, & -H_{\mathrm{sml}}(x) < z < 0, \\
S_{\mathrm{int}}, & \overline{\eta}_b(x) + H_{\mathrm{bbl}}(x) < z < -H_{\mathrm{sml}}, \\
\widetilde{S}_{\mathrm{bbl}}, & \overline{\eta}_b(x) < z < \overline{\eta}_b(x) + H_{\mathrm{bbl}}(x).
\end{cases}
\tag{27}
$$

Thus $S_{\mathrm{gm}}$ and $S_{\mathrm{iso}}$ are identical everywhere above the BBL. The need for a distinction within the BBL is explained below in Sections 2.3.2.1 and 2.3.2.2.

**2.3.2.1   Surface Mixed Layer**

We now discuss the formulation of $S_{\mathrm{sml}}$, the effective isopycnal slope in the surface mixed layer. Following Ferrari et al. (2008), we construct $S_{\mathrm{sml}}$ in a way that avoids singularities due to the vanishingly small vertical buoyancy gradients, and thus near-infinite isopycnal slopes, that occur in the mixed layer. This is achieved by using the vertical buoyancy gradient at the base of the mixed layer to define the effective slope as

$S_{\mathrm{sml}} = -G_{\mathrm{sml}}(\sigma_{\mathrm{sml}}) \dfrac{\partial_x \overline{b}}{\partial_z \overline{b}\big|_{z=-H_{\mathrm{sml}}}},$
$\tag{28}$

where $\sigma_{\mathrm{sml}} = -z/H_{\mathrm{sml}}$ is a dimensionless vertical coordinate for the SML, as in Section 2.3.1. The corresponding eddy streamfunction (21) is identical to that of Ferrari et al. (2008),

$$
\psi^{\star} = -\kappa_{\mathrm{gm}} G_{\mathrm{sml}}(\sigma_{\mathrm{sml}}) \frac{\partial_x \overline{b}}{\partial_z \overline{b}\big|_{z=-H_{\mathrm{sml}}}}, \qquad z \geq -H_{\mathrm{sml}}.
\tag{29}
$$





The structure function $G_{\mathrm{sml}}(z)$ is required to enforce continuity of the vertical tracer fluxes and flux divergences at the surface and at the base of the mixed layer. For example, (23) requires that $G_{\mathrm{sml}}$ vanish at the surface:

$$G_{\mathrm{sml}}(0) = 0. \tag{30}$$

We further require that the eddy streamfunction and eddy residual tracer fluxes be continuous at the base of the SML, *i.e.* that $S_{\mathrm{sml}} = S_{\mathrm{int}}$, which requires that

$$G_{\mathrm{sml}}(1) = 1. \tag{31}$$

Finally, we require continuity of the divergence of the eddy tracer flux in order to avoid producing singularities at the SML base. The zonal and vertical components of the eddy tracer flux are

$$\overline{u'c'} = \kappa_{\mathrm{gm}} S_{\mathrm{gm}} \frac{\partial \overline{c}}{\partial z} - \kappa_{\mathrm{iso}} \left( \frac{\partial \overline{c}}{\partial x} + S_{\mathrm{iso}} \frac{\partial \overline{c}}{\partial z} \right), \tag{32a}$$

$$\overline{w'c'} = -\kappa_{\mathrm{gm}} S_{\mathrm{gm}} \frac{\partial \overline{c}}{\partial x} - \kappa_{\mathrm{iso}} S_{\mathrm{iso}} \left( \frac{\partial \overline{c}}{\partial x} + S_{\mathrm{iso}} \frac{\partial \overline{c}}{\partial z} \right). \tag{32b}$$

It may be shown that continuity of $\nabla \cdot \overline{\mathbf{u}'c'}$ across $z = -H_{\mathrm{sml}}$ is guaranteed if

$$\left. \frac{\partial S_{\mathrm{sml}}}{\partial z} \right|_{z=-H_{\mathrm{sml}}^+} = \left. \frac{\partial S_{\mathrm{int}}}{\partial z} \right|_{z=-H_{\mathrm{sml}}^-} \implies G'_{\mathrm{sml}}(1) = \frac{H_{\mathrm{sml}}}{\lambda_{\mathrm{sml}}} \tag{33}$$

where $\lambda_{\mathrm{sml}} = \partial_{zz}\overline{b}/\partial_z\overline{b}|_{z=-H_{\mathrm{sml}}}$ is a vertical lengthscale for eddy motions at the base of the mixed layer.

The simplest form for $G_{\mathrm{sml}}(z)$ that satisfies conditions (30), (31) and (33) is a quadratic function of depth,

$$G_{\mathrm{sml}}(\sigma_{\mathrm{sml}}) = -\left( 1 - \frac{H_{\mathrm{sml}}}{\lambda_{\mathrm{sml}}} \right) \sigma_{\mathrm{sml}}^2 + \left( 2 - \frac{H_{\mathrm{sml}}}{\lambda_{\mathrm{sml}}} \right) \sigma_{\mathrm{sml}}. \tag{34}$$

Equation (34) is currently implemented in MAMEBUS. A more sophisticated form of $G_{\mathrm{sml}}$ that arguably has stronger physical motivation is given by Ferrari et al. (2008). They split the SML into a true mixed layer, in which $G_{\mathrm{sml}}$ varies linearly (and so the eddy velocity is approximately uniform), overlying a transition layer, in which $G_{\mathrm{sml}}(\sigma_{\mathrm{sml}})$ varies quadratically.

#### 2.3.2.2 Bottom boundary layer

The scheme described above for the SML relies on the fact that the ocean surface is approximately flat, which allows the same effective slopes $S_{\mathrm{sml}}$ to be used for $S_{\mathrm{gm}}$ and $S_{\mathrm{iso}}$. The sloping sea floor requires separate BBL slopes, $S_{\mathrm{bbl}}$ and $\widetilde{S}_{\mathrm{bbl}}$, and structure functions, $G_{\mathrm{bbl}}$ and $\widetilde{G}_{\mathrm{bbl}}$ to satisfy the required conditions of no volume nor tracer flux through the boundary, *i.e.* (23) and (26).

Analogous to the SML, we define the effective slope $S_{\mathrm{bbl}}$ as

$$S_{\mathrm{bbl}} = -G_{\mathrm{bbl}}(\sigma_{\mathrm{bbl}}) \frac{\partial_x \overline{b}}{\partial_z \overline{b}|_{z=\eta_b+H_{\mathrm{bbl}}}}, \tag{35}$$

where $\sigma_{\mathrm{bbl}} = (z - \eta_b(x))/H_{\mathrm{bbl}}(x)$ is the BBL vertical coordinate, as in Section 2.3.1. The eddy streamfunction in the BBL is therefore

$$\psi^\star = -\kappa_{\mathrm{gm}} G_{\mathrm{bbl}}(\sigma_{\mathrm{bbl}}) \frac{\partial_x \overline{b}}{\partial_z \overline{b}|_{z=\eta_b+H_{\mathrm{bbl}}}}, \qquad z \le \eta_b + H_{\mathrm{bbl}}. \tag{36}$$





To satisfy the condition of zero volume flux through the sea floor, (23), the effective slope must vanish at $z = \eta_b(x)$, which requires

$$G_{\mathrm{bbl}}(0) = 0. \tag{37}$$

To ensure continuity of the eddy streamfunction at the top of the BBL, we require that $S_{\mathrm{bbl}}$ approach $S_{\mathrm{int}}$, *i.e.*

$$G_{\mathrm{bbl}}(1) = 1. \tag{38}$$

Finally, to ensure continuity of the eddy bolus velocity, we require that the gradient of $S_{\mathrm{gm}}$ be continuous at $z = \eta_b + H_{\mathrm{bbl}}$. This imposes a constraint analogous to (33) on $G_{\mathrm{bbl}}$,

$$G'_{\mathrm{bbl}}(1) = -\frac{H_{\mathrm{bbl}}}{\lambda_{\mathrm{bbl}}}, \tag{39}$$

where $\lambda_{\mathrm{bbl}} = \bar{b}_{zz}/\bar{b}_z\big|_{z=\bar{\eta}_b+H_{\mathrm{bbl}}}$ is a vertical lengthscale for eddies at the top of the BBL. To satisfy (37)–(39), we select a
quadratic form for the structure function $G_{\mathrm{bbl}}(\sigma_{\mathrm{bbl}})$,

$$G_{\mathrm{bbl}}(\sigma_{\mathrm{bbl}}) = -\left(1 + \frac{H_{\mathrm{bbl}}}{\lambda_{\mathrm{bbl}}}\right)\sigma_{\mathrm{bbl}}^2 + \left(2 + \frac{H_{\mathrm{bbl}}}{\lambda_{\mathrm{bbl}}}\right)\sigma_{\mathrm{bbl}}. \tag{40}$$

However, the effective slope $S_{\mathrm{bbl}}$ can no longer be used to define $S_{\mathrm{iso}}$ in the BBL: (26) requires that the effective slope be aligned with the bottom slope at the sea floor, $S_b = \partial_x \bar{\eta}_b$ at $z = \bar{\eta}_b$. We must therefore employ a modified effective slope $\widetilde{S}_{\mathrm{bbl}}$ in the isopycnal mixing operator, as expressed in Equation (27). We define $\widetilde{S}_{\mathrm{bbl}}$ as

$$\widetilde{S}_{\mathrm{bbl}} = S_{\mathrm{bbl}} + \left(1 - \widetilde{G}_{\mathrm{bbl}}(z)\right) S_b, \tag{41}$$

where $\widetilde{G}_{\mathrm{bbl}}(\sigma_{\mathrm{bbl}})$ is a modified structure function that also vanishes at the ocean bed,

$$\widetilde{G}_{\mathrm{bbl}}(0) = 0. \tag{42}$$

Continuity of the eddy tracer fluxes at the top of the BBL requires that

$$\widetilde{G}_{\mathrm{bbl}}(1) = 1. \tag{43}$$

Finally, continuity of the eddy flux divergence is enforced by

$$\left.\frac{\partial \widetilde{G}_{\mathrm{bbl}}}{\partial z}\right|_{z=\bar{\eta}_b+H_{\mathrm{bbl}}} = 0. \tag{44}$$

To satisfy (42)–(44), we select a quadratic form for the structure function $\widetilde{G}_{\mathrm{bbl}}(\sigma_{\mathrm{bbl}})$,

$$\widetilde{G}_{\mathrm{bbl}}(\sigma_{\mathrm{bbl}}) = \sigma_{\mathrm{bbl}}(2 - \sigma_{\mathrm{bbl}}). \tag{45}$$





## 2.4  Biogeochemical Model Formulation

The current biogeochemical model implemented in MAMEBUS is an NPZD (nutrient, phytoplankton, zooplankton, and detritus) model. This NPZD model is modeled after the size-structured AstroCAT (Banas, 2011) and Darwin models (Ward et al., 2012). For the purpose of this paper, we reduced the size structured ecosystem model to a single phytoplankton and zooplankton size classes, while preserving the option to run multiple size classes in future versions of the model. We also includes a detritus variable, which allows for sinking and export of organic matter away from the euphotic zone, and redistribution of nutrients in the water column.

The biogeochemical equations in MAMEBUS are formulated similarly to previous NPZD models, but cast in terms of the meridionally-averaged nutrient, phytoplankton, zooplankton and detritus concentrations. We neglect additional terms that would be introduced by first formulating the equations and then taking the meridional average, *e.g.* covariances of the type $\overline{P'Z'}$. This assumption is partially predicated on the idea that zonal gradients in biogeochemical tracers (e.g. nutrients and chlorophyll) are much stronger than meridional gradients, as supported by observations and models (Fiechter et al., 2018). For example, Venegas et al. (2008) show that average chlorophyll concentrations during the upwelling season vary approximately two-fold in the Northern California Current System, whereas observations from CalCOFI (Figure 7) show variations by an order of magnitude between nearshore and offshore stations. Alongshore gradients in chlorophyll are observed along the coast, where they are driven by wind and topographic variations; however they are generally much smaller than the gradient between the coast and the offshore region (Fiechter et al., 2018). We recognize that this is a simplification of the true variability in EBUSs, but we consider it appropriate on average over the entire upwelling system, in particular within the idealized MAMEBUS framework, and plan to reassess it in future work.

We drop the bar notation indicating a meridional average for this section, with the understanding that all variables denote meridionally-averaged quantities. In the following, we include size dependent uptake and grazing, along with variable sinking speeds for detritus, to retain essential size-dependent biogeochemical interactions and export fluxes. This will facilitate a future introduction of multiple size classes in the model. All variables and coefficients are given in Table 1. We note that all of the parameter values and equations described below measure time in days, whereas more generally MAMEBUS measures time in seconds; appropriate conversions are made in the model code to ensure dimensional consistency. The main conservation equations for biogeochemical tracers are:

$$\left.\frac{\partial N}{\partial t}\right|_{\text{bio}} = -\mathcal{U}(N, I, T, P) + \mathcal{R}(D), \tag{46a}$$

$$\left.\frac{\partial P}{\partial t}\right|_{\text{bio}} = \mathcal{U}(N, I, T, P) - \mathcal{G}(P, Z) - \mathcal{M}(P), \tag{46b}$$

$$\left.\frac{\partial Z}{\partial t}\right|_{\text{bio}} = \lambda \mathcal{G}(P, Z) - \mathcal{M}(Z), \tag{46c}$$

$$\left.\frac{\partial D}{\partial t}\right|_{\text{bio}} = \mathcal{M}(P) + \mathcal{M}(Z) + (1 - \lambda)\mathcal{G}(P, Z) - \frac{\partial}{\partial z} w_{\text{sink}} D - \mathcal{R}(D), \tag{46d}$$





**Table 1.** Parameters and values used in the ecosystem model implemented in MAMEBUS. Coefficients without explicit references are chosen by the user.

| Parameter | Value | Units | Description | Reference |
|---|---|---|---|---|
| $I_p$ | 0.45 | | Fraction of light available for photosynthesis (PAR) | Moore et al. (2001) |
| $k_c$ | 0.01 | 1/(mmol N)m | Absorption coefficient for photosynthesis | Moore et al. (2001) |
| $k_p$ | 3 | mmol N/m$^3$ | Half saturation coefficient for phytoplankton grazing | Banas (2011) |
| $k_w$ | 0.04 | 1/m | Absorption coeffcient for water | Moore et al. (2001) |
| $\Delta\ell$ | 0.25 | $\log_{10}\mu$m | Width of grazing profile | Banas (2011) |
| $\ell_p$ | 5 | $\mu$m | Length (ESD) of phytoplankton cell | |
| $\ell_z$ | 10 | $\mu$m | Length (ESD) of zooplankton cell | |
| $\lambda$ | 0.33 | | Biomass assimilation efficiency | |
| $Q_{\text{sw}}$ | 340 | W/m$^2$ | Surface irradiance | Moore et al. (2001) |
| $r_T$ | 0.05 | 1/$^\circ$C | Temperature dependence of nutrient uptake | Ward et al. (2012) |
| $r_{\text{remin}}$ | 0.04 | 1/d | Remineralization rate | Ward et al. (2012) |
| $T_0$ | 10 | $^o$C | Reference temperature | |
| $\mu_p$ | 0.02 | | Phytoplankton mortality as a fraction of growth rate | Banas (2011) |
| $\mu_z$ | 0.97 – 12.57 | m$^3$/(mmol N d) | Density dependent zooplankton mortality | Edwards and Bees (2001) |
| $w_{\text{sink}}$ | 10 | m/d | Sinking speed of detritus | |

where $T$ ($^\circ$C) is the model temperature, $I$ (W/m$^2$) is the local irradiance profile, $N$ (mmol N/m$^3$) is nitrate concentration, $P$ (mmol N/m$^3$) is phytoplankton concentration, $Z$ (mmol N/m$^3$) is zooplankton concentration, and $D$ (mmol N/m$^3$) is the detritus concentration. The terms on the right-hand sides of (46a)–(46d) are explained in the following subsections.

### 2.4.1 Nutrient Uptake

5  Common controls on phytoplankton population are bottom-up limitation (i.e. nutrient control), and top-down grazing by zooplankton (Sarmiento and Gruber, 2006). We formulate bottom-up controls using typical choices for light- and temperature-dependent terms, and Michaelis-Menten uptake (Sarmiento and Gruber, 2006). The functional form of uptake is given by:

$$\mathcal{U}(N, I, T, P) = \varphi(I)\varphi(T)U^{\text{max}}\frac{N}{N + k_N}P, \tag{47}$$

10  where $\varphi(I)$ and $\varphi(T)$ are light and temperature limiting functions, respectively. The light attenuation is modeled by integrating the Beer-Lambert Law, following Moore et al. (2001),

$$\frac{\partial I(z)}{\partial z} = -k_{\text{par}}I(z), \quad \text{where} \quad I_0 = I(z = 0) = Q_{sw}I_p, \tag{48a}$$

$$k_{\text{par}} = k_w + P \cdot k_c, \tag{48b}$$



**Table 2.** Parameters and values used in the ecosystem model implemented in MAMEBUS. Coefficients without explicit references are chosen by the user.

| Parameter | Value | Units | Description | Reference |
|-----------|-------|-------|-------------|-----------|
| $a_u$ | 2.6 | 1/d | Uptake rate | Tang (1995) |
| $b_u$ | -0.45 | | Scaling parameter for uptake | Tang (1995) |
| $a_g$ | 26 | 1/d | Grazing rate | Hansen et al. (1994) |
| $b_g$ | -0.4 | | Scaling parameter for grazing | Hansen et al. (1994) |
| $a_o$ | 0.65 | $\mu$m | Optimal predator-prey length scale | Hansen et al. (1994) |
| $b_o$ | 0.56 | | Scaling parameter for optimal predator-prey interaction | Hansen et al. (1994) |

and the light-dependent uptake function is modeled following Sarmiento and Gruber (2006),

$$\varphi(I) = \frac{I(z)}{\sqrt{I_0^2 + I(z)^2}}. \tag{49}$$

The temperature component of the uptake function is,

$$\varphi(T) = e^{-r_T(T-T_0)}. \tag{50}$$

The maximum uptake rate is an allometric relationship defined as,

$$U^{\mathrm{max}} = a_u \left(\frac{\ell_p}{\ell_0}\right)^{b_u}, \tag{51}$$

where $\ell_p$ is the user-determined phytoplankton size expressed as equivalent spherical diameter (ESD), and $\ell_0 = 1\mu$m is a normalized length scale, with all allometrically defined variable listed in Table 2. While there are other options for the bases of these allometric relationships outlined in this section, (eg. cell volume), we make the decision to use ESD as a measure of cell

size. Finally, the half saturation coefficient is $k_N = 0.1$ mmol N/m$^3$.

### 2.4.2 Grazing

Top-down processes are represented by zooplankton grazing on phytoplankton. Andersen et al. (2016) noted that there is an optimal length scale for active predation and grazing, as a strategic trade-off for optimal biomass assimilation. We make the assumption that the biomass assimilation of phytoplankton by zooplankton also follows Michaelis-Menten dynamics, then the

functional form of grazing is given by

$$\mathcal{G}(P,Z) = G^{\mathrm{max}}\frac{\vartheta P}{k_P + \vartheta P}Z, \tag{52}$$

where the maximum grazing rate is defined by an allometric relationship defined as,

$$G^{\mathrm{max}} = a_g \left(\frac{\ell_z}{\ell_0}\right)^{b_g}. \tag{53}$$





where d$^{-1}$ represents a "per day" quantity. We define a Gaussian distribution about an optimal grazing length-scale following Banas (2011),

$$\vartheta = \exp\left(-\frac{\log_{10}(\ell_p) - \log_{10}(\ell_{\text{opt}})}{\Delta\ell}\right), \tag{54}$$

where $\Delta\ell$ sets the width of the optimal grazing profile, and defines a band of grazing about the optimal prey size, $\ell_{\text{opt}}$. By
allowing for a variable band of grazing, we are able to control the assimilation efficiency of phytoplankton by zooplankton through direct preferential grazing. Accordingly, we model the optimal prey size based on a preferential grazing profile centered about an optimal predator-prey length scale,

$$\ell_{\text{opt}} = a_o \left(\frac{\ell_z}{\ell_0}\right)^{b_o}. \tag{55}$$

### 2.4.3 Mortality

Mortality closure terms often set important internal dynamics in ecosystem models (Poulin and Franks, 2010). While linear mortality terms are generally used for phytoplankton, zooplankton mortality is often modeled via a quadratic term to avoid unrealistic oscillations and stabilize the solution (Poulin and Franks, 2010). The quadratic mortality term may be rationalized as a representation of mixotrophic grazing, zooplankton self-grazing and higher order grazing in NPZD models (Raick et al., 2006). Therefore, we model phytoplankton mortality as,

$$\mathcal{M}(P) = \mu_p U^{\text{max}} P, \tag{56}$$

and zooplankton mortality as,

$$\mathcal{M}(Z) = \mu_z Z^2. \tag{57}$$

### 2.4.4 Remineralization and Particle Sinking

Sinking particles are an essential component of the vertical transport of nutrients from the surface to the deep ocean (Sarmiento
and Gruber, 2006). Once particles sink past the euphotic zone, they are remineralized and returned to the subsurface nutrient pool. In this model, we represent remineralization processes via a linear rate, *i.e.*:

$$\mathcal{R}(D) = r_{\text{remin}} D. \tag{58}$$

where $r_{\text{remin}}$ is the specific remineralization rate.

Particles sink at a constant average speed in the water column, following Equation (46d). At the bottom boundary we impose
zero sinking flux, *i.e.* $w_{\text{sink}} = 0$ at $z = \overline{\eta}_b(x)$. Thus any nutrients that sink to the sea floor as detritus must remineralize there. This allows for redistribution of nutrients by mixing within the bottom boundary layer, diffusion into the interior, and transport via upwelling onto the shelf.



## 2.5 Non-conservative Terms

In this section we describe the treatment of all non-conservative terms in the tracer evolution equation. MAMEBUS allows arbitrary restoring of all tracers, which may be used, for example, to impose offshore boundary conditions or to impose restoring at the sea surface. Fixed fluxes of all tracers may also be imposed through the surface. More precisely, we formulate the non-

conservative tracer tendency as

$$\left.\frac{\partial \overline{c}}{\partial t}\right|_{\mathrm{nct}} = \left.\frac{\partial \overline{c}}{\partial t}\right|_{\mathrm{restore}} + \left.\frac{\partial \overline{c}}{\partial t}\right|_{\mathrm{flux}}. \tag{59}$$

The restoring and surface flux components of this tendency are discussed separately below.

### 2.5.1 Restoring

The restoring of a tracer is represented as an exponential decay to a prescribed, spatially-varying tracer field, $\overline{c}_r(x,z)$, with

time scale $t_r(x,z)$. The tracer restoring is then formulated as

$$\left.\frac{\partial \overline{c}}{\partial t}\right|_{\mathrm{restore}} = -\frac{\overline{c} - \overline{c}_r}{t_r}. \tag{60}$$

### 2.5.2 Tracer fluxes

Surface fluxes are represented as a tendency in the tracer concentration in the surface gridboxes. For an arbitrary tracer $c$, we formulate the surface flux term as follows:

$$\left.\frac{\partial \overline{c}}{\partial t}\right|_{\mathrm{flux}} = \frac{\partial F_{\mathrm{flux}}^c}{\partial z}, \qquad F_{\mathrm{flux}} = \begin{cases} F_{\mathrm{flux},0}^c, & z = 0, \\ 0, & z < 0. \end{cases} \tag{61}$$

Here $F_{\mathrm{flux},0}^c$ is the downward flux of $c$ (units of $[c]$m/s) at the surface. For the case of buoyancy, the surface flux is imposed as a surface energy flux, $Q_s$ (W/m$^2$), with

$$F_{\mathrm{flux},0}^b = \frac{g\alpha Q_s}{\rho_0 C_p}, \tag{62}$$

where $C_p = 4000\mathrm{J}/^\circ\mathrm{C\ kg}$ is the specific heat capacity.





## 3 MAMEBUSv1.0 Algorithm

In this section we discuss the numerical solution of the model equations presented in Section 2. This entails a recasting of the equations in terrain-following, or "sigma" coordinates (*e.g.* Song and Haidvogel, 1994; Shchepetkin and McWilliams, 2003), followed by the spatial discretization of the equations and algorithms for numerical time stepping.

### 3.1 Formulation in terrain-following coordinates

We solve the model equations presented in Section 2 in a coordinate system that "stretches" in the vertical to follow the shape of the sea floor. Such a coordinate system avoids "steps" in the sea floor that arise, for example, when using geopotential vertical coordinates, and allows fine vertical resolution of the bottom boundary layer (*e.g.* Song and Haidvogel, 1994; Shchepetkin and McWilliams, 2003). Formally, we make a coordinate transformation $(x,z) \rightarrow (x,\sigma)$, where $\sigma$ is a dimensionless vertical

coordinate and is defined such that $\sigma = 0$ at $z = 0$ and $\sigma = -1$ at $z = \overline{\eta}_b(x)$. This transformation requires a relationship between $z$ and $\sigma$ via a transformation function

$$z = \zeta(x,\sigma). \tag{63}$$

For example, a "pure" sigma coordinate corresponds to the choice

$$\zeta(x,\sigma) = -\sigma\overline{h}_b(x), \tag{64}$$

where $\overline{h}_b(x) = -\overline{\eta}_b(x)$ is the merionally-averaged water column thickness. However, this is not necessarily the most practical choice for numerical applications, in which it is useful to focus the vertical resolution over certain depth ranges (especially those close to the top and bottom boundaries of the ocean). MAMEBUS currently implements the UCLA-ROMS (Shchepetkin and McWilliams, 2005) transformation function,

$$\zeta(x,\sigma) = h_b(x) \left[ \frac{h_c\sigma + h_b(x)C(\sigma)}{h_c + h_b(x)} \right]. \tag{65}$$

Here $C(\sigma)$ is the stretching function, defined as

$$C(\sigma) = \begin{cases} \dfrac{\exp\left(\theta_b\tilde{C}(\sigma)\right) - 1}{1 - \exp(-\theta_b)}, & \theta_b > 0, \\ \tilde{C}(\sigma), & \theta_b \leq 0, \end{cases} \tag{66}$$

where

$$\tilde{C}(\sigma) = \begin{cases} \dfrac{1 - \cosh(\theta_s\sigma)}{\cosh(\theta_s) - 1}, & \theta_s > 0, \\ -\sigma^2, & \theta_s \leq 0. \end{cases} \tag{67}$$

Here $C$ and $\tilde{C}$ are the bottom and surface components of the stretching function, respectively. The parameters $\theta_s \in [0,10]$

and $\theta_b \in [0,4]$ are surface and bottom stretching parameters; larger values cause the near-surface and near-bottom portions of





the domain to occupy larger fraction of $\sigma$-space. The parameter $h_c$ defines a surface layer thickness, in which the coordinate system is approximately aligned with geopotentials, provided that $h_b \gg h_c$.

We now write the physical tracer evolution Equation (9) in $\sigma$-coordinates. For a given function $f = f(x, z(x, \sigma))$, we can write derivatives with respect to $x$ and $\sigma$ as

$$\left.\frac{\partial f}{\partial x}\right|_\sigma = \left.\frac{\partial f}{\partial x}\right|_z + \frac{\partial \zeta}{\partial x}\left.\frac{\partial f}{\partial z}\right|_x, \tag{68a}$$

$$\left.\frac{\partial f}{\partial \sigma}\right|_x = \frac{\partial \zeta}{\partial \sigma}\left.\frac{\partial f}{\partial z}\right|_x. \tag{68b}$$

Using these identities, we may write the divergence of an arbitrary vector $\mathbf{F}$, with components $F^{(x)}$ and $F^{(z)}$ in the $\hat{\mathbf{x}}$ and $\hat{\mathbf{z}}$ directions, respectively, as

$$\nabla \cdot \mathbf{F} = \left.\frac{\partial}{\partial x}\right|_z F^{(x)} + \left.\frac{\partial}{\partial z}\right|_x F^{(z)} = \zeta_\sigma^{-1}\left.\frac{\partial}{\partial x}\right|_\sigma \left(\zeta_\sigma F^{(x)}\right) + \zeta_\sigma^{-1}\left.\frac{\partial}{\partial \sigma}\right|_x \left(F^{(z)} - \zeta_x F^{(x)}\right). \tag{69}$$

Equation (69), combined with the definition of the mean streamfunction (12), allows us to write the mean advection term in (9) as

$$\nabla \cdot (\overline{\mathbf{u}}\overline{c}) = \nabla \cdot \left(-\overline{c}\left.\frac{\partial \overline{\psi}}{\partial z}\right|_x, \overline{c}\left.\frac{\partial \overline{\psi}}{\partial x}\right|_z\right) = \zeta_\sigma^{-1}\left.\frac{\partial}{\partial x}\right|_\sigma \left(-\overline{c}\left.\frac{\partial \overline{\psi}}{\partial \sigma}\right|_x\right) + \zeta_\sigma^{-1}\left.\frac{\partial}{\partial \sigma}\right|_x \left(\overline{c}\left.\frac{\partial \overline{\psi}}{\partial x}\right|_\sigma\right). \tag{70}$$

An analogous expression may be obtained for the eddy advection term, $\nabla \cdot (\mathbf{u}^\star \overline{c})$, in (9), using the definition (21) of the eddy streamfunction. Next, we apply (69) to the isopycnal mixing operator, defined by (25), in Equation (9) to obtain

$$\nabla \cdot \left(\kappa_{\mathrm{iso}}\nabla_\| \overline{c}\right) = \zeta_\sigma^{-1}\left.\frac{\partial}{\partial x}\right|_\sigma \left[\zeta_\sigma \kappa_{\mathrm{iso}}\left(\left.\frac{\partial \overline{c}}{\partial x}\right|_\sigma + (S_{\mathrm{iso}} - S_\sigma)\zeta_\sigma^{-1}\left.\frac{\partial \overline{c}}{\partial \sigma}\right|_x\right)\right]$$
$$+ \zeta_\sigma^{-1}\left.\frac{\partial}{\partial \sigma}\right|_x \left[\kappa_{\mathrm{iso}}(S_{\mathrm{iso}} - S_\sigma)\left(\left.\frac{\partial \overline{c}}{\partial x}\right|_\sigma + (S_{\mathrm{iso}} - S_\sigma)\zeta_\sigma^{-1}\left.\frac{\partial \overline{c}}{\partial \sigma}\right|_x\right)\right], \tag{71}$$

where $S_\sigma = \zeta_x$ is the slope of surfaces of constant $\sigma$ in $x/z$ space, *i.e.* the slope of the $\sigma$-coordinate grid lines. Thus the isopycnal mixing operator is essentially just modified by subtracting $S_\sigma$ from $S_{\mathrm{iso}}$ to obtain the mixing slope *relative to the slope of the $\sigma$-coordinate grid*. Over most of the water column $S_{\mathrm{iso}}$ is equal to the isopycnal slope $S_{\mathrm{int}}$, given by

$$S_{\mathrm{int}} = -\frac{\left.\frac{\partial \overline{b}}{\partial x}\right|_z}{\left.\frac{\partial \overline{b}}{\partial z}\right|_x} = -\frac{\left.\frac{\partial \overline{b}}{\partial x}\right|_\sigma - S_\sigma\left.\frac{\partial \overline{b}}{\partial z}\right|_x}{\left.\frac{\partial \overline{b}}{\partial z}\right|_x} = -\frac{\left.\frac{\partial \overline{b}}{\partial x}\right|_\sigma}{\zeta_\sigma^{-1}\left.\frac{\partial \overline{b}}{\partial \sigma}\right|_x} + S_\sigma. \tag{72}$$

Thus, the quantity $S_{\mathrm{int}} - S_\sigma$ can actually be computed more directly than the true isopycnal slope, as

$$S_{\mathrm{int}} - S_\sigma = -\frac{\left.\frac{\partial \overline{b}}{\partial x}\right|_\sigma}{\zeta_\sigma^{-1}\left.\frac{\partial \overline{b}}{\partial \sigma}\right|_x}. \tag{73}$$

Finally, the $\sigma$-coordinate transformation of the vertical (quasi-diapycnal) mixing operator is

$$\left.\frac{\partial}{\partial z}\right|_x \left(\kappa_{\mathrm{dia}}\left.\frac{\partial \overline{c}}{\partial z}\right|_x\right) = \zeta_\sigma^{-1}\left.\frac{\partial}{\partial \sigma}\right|_x \left(\kappa_{\mathrm{dia}}\zeta_\sigma^{-1}\left.\frac{\partial \overline{c}}{\partial \sigma}\right|_x\right). \tag{74}$$





To summarize, we write (9) in $\sigma$ coordinates as

$$\left.\frac{\partial \overline{c}}{\partial t}\right|_{\text{phys}} = G_{\text{adv}} + G_{\text{iso}} + G_{\text{dia}} + G_{\text{lat}}, \tag{75}$$

where the tendency terms are

$$G_{\text{adv}} = -\zeta_\sigma^{-1} \left.\frac{\partial}{\partial x}\right|_\sigma \left[ \zeta_\sigma \left( -\overline{c}\zeta_\sigma^{-1} \left.\frac{\partial \psi^\dagger}{\partial \sigma}\right|_x \right) \right] - \zeta_\sigma^{-1} \left.\frac{\partial}{\partial \sigma}\right|_x \left( \overline{c} \left.\frac{\partial \psi^\dagger}{\partial x}\right|_\sigma \right), \tag{76a}$$

$$G_{\text{iso}} = \zeta_\sigma^{-1} \left.\frac{\partial}{\partial x}\right|_\sigma \left[ \zeta_\sigma \kappa_{\text{iso}} \left( \left.\frac{\partial \overline{c}}{\partial x}\right|_\sigma + (S_{\text{iso}} - S_\sigma)\zeta_\sigma^{-1} \left.\frac{\partial \overline{c}}{\partial \sigma}\right|_x \right) \right],$$

$$\qquad + \zeta_\sigma^{-1} \left.\frac{\partial}{\partial \sigma}\right|_x \left[ \kappa_{\text{iso}}(S_{\text{iso}} - S_\sigma) \left( \left.\frac{\partial \overline{c}}{\partial x}\right|_\sigma + (S_{\text{iso}} - S_\sigma)\zeta_\sigma^{-1} \left.\frac{\partial \overline{c}}{\partial \sigma}\right|_x \right) \right], \tag{76b}$$

$$G_{\text{dia}} = \zeta_\sigma^{-1} \left.\frac{\partial}{\partial \sigma}\right|_x \left( \kappa_{\text{dia}}\zeta_\sigma^{-1} \left.\frac{\partial \overline{c}}{\partial \sigma}\right|_x \right) \tag{76c}$$

$$G_{\text{lat}} = -\frac{\overline{v}}{L_y} \left[ c \right]_0^{L_y}. \tag{76d}$$

Here we define

$$\psi^\dagger = \overline{\psi} + \psi^\star \tag{77}$$

as the total advective or "residual" streamfunction (Plumb and Ferrari, 2005b), and we have added factors of $\zeta_\sigma$ in (76a) so that the fluxes can be directly identified with the zonal velocity, $u^\dagger = -\zeta_\sigma^{-1}\partial\psi^\dagger/\partial\sigma|_x$, and the dia-$\sigma$ velocity, $\varpi^\dagger = \partial\psi^\dagger/\partial x|_\sigma$. Note also that every derivative with respect to $\sigma$ is multiplied by $\zeta_\sigma^{-1}$, and that their product $\zeta_\sigma^{-1}\partial_\sigma$ is equivalent to a derivative with respect to $z$. This allows us to simplify the numerical discretization by avoiding explicit references to $\sigma$ coordinates, and computing these derivatives via finite differencing in $z$ coordinates.

## 3.2 Spatial discretization of the tracer evolution equation

We solve (9) using the slope-limited finite-volume scheme of Kurganov and Tadmor (2000) for systems of conservation laws. We divide the domain into a grid of $N_x$ by $N_\zeta$ cells, with uniform side lengths $\Delta_x$ and $\Delta_\zeta$ in $x/\zeta$ space, as shown in Fig. 2. We store the cell-averaged value of $\overline{c}$ at the center of the $(j,k)^{\text{th}}$ grid cell, which we denote as $\overline{c}_{j,k}(t)$. The mean, eddy and residual streamfunctions are most naturally defined at the cell corners, as this allows a straightforward calculation of the residual velocities at the cell edges,

$$u^\dagger_{j+1/2,k} = -\frac{\psi^\dagger_{j+1/2,k+1/2} - \psi^\dagger_{j+1/2,k-1/2}}{(\Delta_z)_{j+1/2,k}}, \tag{78a}$$

$$\varpi^\dagger_{j,k+1/2} = \frac{\psi^\dagger_{j+1/2,k+1/2} - \psi^\dagger_{j-1/2,k+1/2}}{\Delta_x}. \tag{78b}$$

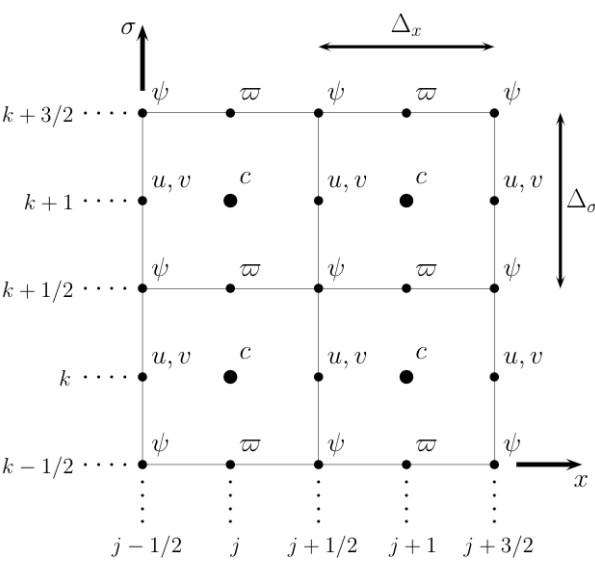

**Figure 2.** Illustration of the numerical grid used to compute solutions to the model equations.

Here we use $\Delta_z$ as a shorthand for the spatially-varying vertical grid spacing, defined as a centered difference between adjacent grid points. The vertical grid spacing is defined for all cell centers, corners and faces,

$$(\Delta_z)_{j,k} = z_{j,k+1/2} - z_{j,k-1/2}, \tag{78c}$$

$$(\Delta_z)_{j+1/2,k} = z_{j+1/2,k+1/2} - z_{j+1/2,k-1/2}, \tag{78d}$$

$$(\Delta_z)_{j,k+1/2} = z_{j,k+1} - z_{j,k}, \tag{78e}$$

$$(\Delta_z)_{j+1/2,k+1/2} = z_{j+1/2,k+1} - z_{j+1/2,k}, \tag{78f}$$

where $z_{j+1/2,k+1/2}$ denotes the physical elevation of each gridpoint. Note again that $\varpi$ is the velocity normal to the upper and lower faces of the grid cell, and so differs slightly from the true vertical velocity $w$.

To compute the advective tendency, $G_{\text{adv}}$, the Kurganov and Tadmor (2000) scheme requires a linear interpolation of $\bar{c}$ over each grid cell. The linear slopes in the $x$ and $\zeta$ directions around $\bar{c}_{j,k}(t)$ are calculated via slope-limited finite differences between $\bar{c}_{j,k}(t)$ and its adjacent gridpoints,

$$(\partial_x \bar{c})_{j,k} = \text{minmod}\left( \theta \frac{\bar{c}_{j+1,k} - \bar{c}_{j,k}}{\Delta_x}, \frac{\bar{c}_{j+1,k} - \bar{c}_{j-1,k}}{2\Delta_x}, \theta \frac{\bar{c}_{j,k} - \bar{c}_{j-1,k}}{\Delta_x} \right), \tag{79}$$

$$(\partial_z \bar{c})_{j,k} = \text{minmod}\left( \theta \frac{\bar{c}_{j,k+1} - \bar{c}_{j,k}}{(\Delta_z)_{j,k+1/2}}, \frac{\bar{c}_{j,k+1} - \bar{c}_{j,k-1}}{z_{j,k+1} - z_{j,k-1}}, \theta \frac{\bar{c}_{j,k} - \bar{c}_{j,k-1}}{(\Delta_z)_{j,k-1/2}} \right), \tag{80}$$

with parameter $1 < \sigma < 2$. The minmod function evaluates to zero if its arguments have differing signs, and otherwise evaluates to its argument with smallest modulus. The cell center estimates of the derivatives are then used to construct two different





estimates of $\bar{c}$ at each cell faces via

$$\bar{c}^{(-)}_{j+1/2,k} = \bar{c}_{j,k} + \tfrac{1}{2}\Delta_x(\partial_x\bar{c})_{j,k}, \tag{81a}$$

$$\bar{c}^{(+)}_{j+1/2,k} = \bar{c}_{j+1,k} - \tfrac{1}{2}\Delta_x(\partial_x\bar{c})_{j+1,k}, \tag{81b}$$

$$\bar{c}^{(-)}_{j,k+1/2} = \bar{c}_{j,k} + (z_{j,k+1/2} - z_{j,k})(\partial_z\bar{c})_{j,k}, \tag{81c}$$

$$\bar{c}^{(+)}_{j,k+1/2} = \bar{c}_{j,k} - (z_{j,k+1} - z_{j,k+1/2})(\partial_z\bar{c})_{j,k+1/2}. \tag{81d}$$

Finally, advective fluxes are determined at the cell faces using an estimate of the maximum propagation speed in the system, which in our case is simply the residual velocity, and thus the fluxes reduce to an upwind approximation

$$F^{(u)}_{j+1/2,k} = \frac{1}{2}(\Delta_z)_{j+1/2,k}\left[u^{\dagger}_{j+1/2,k}\left(\bar{c}^{(+)}_{j+1/2,k} + \bar{c}^{(-)}_{j+1/2,k}\right) - |u^{\dagger}_{j+1/2,k}|\left(\bar{c}^{(+)}_{j+1/2,k} - \bar{c}^{(-)}_{j+1/2,k}\right)\right], \tag{82a}$$

$$F^{(\varpi)}_{j,k+1/2} = \frac{1}{2}\left[\varpi^{\dagger}_{j,k+1/2}\left(\bar{c}^{(+)}_{j,k+1/2} + \bar{c}^{(-)}_{j,k+1/2}\right) - |\varpi^{\dagger}_{j,k+1/2}|\left(\bar{c}^{(+)}_{j,k+1/2} - \bar{c}^{(-)}_{j,k+1/2}\right)\right]. \tag{82b}$$

For this version of the model, the formulation of the Kurganov and Tadmor (2000) scheme considers only the maximum propagation speed of the momentum, **u**, and excludes the internal gravity wave speed which is supported with the momentum calculation in Section 2.2. As a result, this would alter the overall advective fluxes, however, we omit this in the current version of the model and note that the full formulation can be implemented here, but we choose to leave this calculation to be updated in a future version of the model. The advective tendency in $(x, z)$ space is then computed via straightforward finite-differencing of these fluxes,

$$(G_{\mathrm{adv}})_{j,k} = -\frac{F^{(u)}_{j+1/2,k} - F^{(u)}_{j-1/2,k}}{\Delta_x(\Delta_z)_{j,k}} - \frac{F^{(\varpi)}_{j,k+1/2} - F^{(\varpi)}_{j,k-1/2}}{(\Delta_z)_{j,k}} - \frac{F^{v}_{j,k}}{L_y}. \tag{83}$$

The advective discretization (83) requires the residual streamfunction to be known on all grid cell corners, which allows the numerical fluxes to be computed at the cell edges. The mean streamfunction $\overline{\psi}$ is computed from the mean velocity field via (12),

$$\overline{\psi}_{j+1/2,k+1/2} = -\sum_{m=0}^{k}\overline{u}_{j+1/2,k}(\Delta_z)_{j+1/2,k}. \tag{84}$$

The eddy streamfunction (21) depends on the "true" slope (72) of the local $\bar{b}$ contours, which we discretize as

$$(S_{\mathrm{int}})_{j+1/2,k+1/2} = -\frac{(\partial_x\bar{b}|_z)_{j+1/2,k+1/2}}{(\partial_z\bar{b}|_x)_{j+1/2,k+1/2}}. \tag{85}$$

The calculation of the derivatives with respect to $x$ and $z$ is described in Section 3.5. We then construct $\psi^{\star}$ on cell corners as

$$\psi^{\star}_{j+1/2,k+1/2} = (\kappa_{\mathrm{gm}})_{j+1/2,k+1/2}(S_{\mathrm{int}})_{j+1/2,k+1/2}. \tag{86}$$



The tracer tendency due to isopycnal diffusion, $G_{\text{iso}}$, is discretized following the formulation of Kurganov and Tadmor (2000) for parabolic operators.

$$H^{(x)}_{j+1/2,k} = (\Delta_z)_{j+1/2,k}(\kappa_{\text{iso}})_{j+1/2,k}\left[\frac{\overline{c}_{j+1,k} - \overline{c}_{j,k}}{\Delta_x} + (S_{\text{iso}} - S_\sigma)_{j+1/2,k}\frac{(\partial_z\overline{c})_{j,k} + (\partial_z\overline{c})_{j+1,k}}{2}\right], \tag{87a}$$

$$H^{(\sigma)}_{j,k+1/2} = (\kappa_{\text{iso}})_{j,k+1/2}(S_{\text{iso}} - S_\sigma)_{j,k+1/2}\left[\frac{(\partial_x\overline{c})_{j,k} + (\partial_x\overline{c})_{j,k+1}}{2} + (S_{\text{iso}} - S_\sigma)_{j,k+1/2}\frac{\overline{c}_{j,k+1} - \overline{c}_{j,k}}{(\Delta z)_{j,k+1/2}}\right], \tag{87b}$$

where $(\partial_z\overline{c})_{j,k}$, $(\partial_z\overline{c})_{j+1,k}$, $(\partial_x\overline{c})_{j,k}$, and $(\partial_x\overline{c})_{j,k+1}$ are computed via (81a)–(81d). In the interior, the isopycnal diffusion slope $S_{\text{iso}} = S_{\text{int}}$ and is calculated on cell corners via (85) and interpolated to cell faces via

$$(S_{\text{iso}})_{j+1/2,k} = \frac{1}{2}\left((S_{\text{iso}})_{j+1/2,k+1/2} + (S_{\text{iso}})_{j+1/2,k-1/2}\right), \tag{88a}$$

$$(S_{\text{iso}})_{j,k+1/2} = \frac{1}{2}\left((S_{\text{iso}})_{j+1/2,k+1/2} + (S_{\text{iso}})_{j-1/2,k+1/2}\right). \tag{88b}$$

The diffusive tendency is then computed via straightforward finite-differencing of the $H$ fluxes,

$$(G_{\text{iso}})_{j,k} = -\frac{H^{(x)}_{j+1/2,k} - H^{(x)}_{j-1/2,k}}{\Delta_x(\Delta_z)_{j,k}} - \frac{H^{(\sigma)}_{j,k+1/2} - H^{(\sigma)}_{j,k-1/2}}{(\Delta_z)_{j,k}} \tag{89}$$

The tracer tendency due to diapycnal mixing, $G_{\text{dia}}$, is computed implicitly. During each time step, all other physical and biogeochemical tendencies are computed and used to advance $\overline{c}_{j,k}$ forward one time step $\Delta_t$, *i.e.*

$$\overline{c}^\star_{j,k} = \overline{c}^n_{j,k} + \mathcal{F}[\overline{c}^n]. \tag{90}$$

Here $n$ denotes the time step number, and $\overline{c}^\star$ denotes an estimate of $\overline{c}$ at $t + \Delta_t$ (see Section 3.3 for details of the time stepping schemes). The updated tracer concentration is then further modified via the addition of a "correction" due to diapycnal diffusion. At each longitude, or for each $j$, we solve

$$\frac{\overline{c}^{n+1}_{j,k} - \overline{c}^\star_{j,k}}{\Delta_t} = \frac{1}{z_{j,k+1/2} - z_{j,k-1/2}}\left[(\kappa_{\text{dia}})_{j,k+1/2}\frac{\overline{c}^{n+1}_{j,k+1} - \overline{c}^{n+1}_{j,k}}{z_{j,k+1} - z_{j,k}} - (\kappa_{\text{dia}})_{j,k-1/2}\frac{\overline{c}^{n+1}_{j,k} - \overline{c}^{n+1}_{j,k-1}}{z_{j,k} - z_{j,k-1}}\right]. \tag{91}$$

Equation (91) defines a tridiagonal matrix system of algebraic equations for the unknowns $\{\overline{c}^{n+1}_{j,k}|k = 1 \ldots N_z\}$, which is inverted using the Thomas algorithm.

Finally, the meridional advection is discretized via a straightforward upwind advection scheme,

$$(G_{\text{lat}})_{j,k} = \frac{v^{(c)}_{j,k}}{L_y}\left(c_{j,k} - c^u_{j,k}\right), \qquad v^{(c)}_{j,k} = \frac{1}{2}\left(v^\dagger_{j+1/2,k} + v^\dagger_{j-1/2,k}\right) \tag{92}$$

where $v^{(c)}$ denotes the meridional velocity on tracer points and $c^u$ denotes the upstream tracer concentration, defined as

$$c^u_{j.k} = \begin{cases} c^N_{j,k}, & v < 0, \\ c^S_{j,k}, & v > 0. \end{cases} \tag{93}$$

Here $c^N$ and $c^S$ are the tracer concentrations at the northern and southern ends of the domain, respectively. In all of the steps listed above, conditions of zero residual streamfunction and zero normal tracer flux are applied at the domain boundaries. These conditions are imposed by simply setting $\psi^\dagger$ to zero on all boundary points, and by setting the numerical fluxes ($F$, $H$, etc.) to zero on the boundary cell faces.





### 3.3 Temporal Discretization

MAMEBUS evolves the model equations forward in time using Adams-Bashforth methods (*e.g.* Durran, 1991) modified to allow for adaptive time step sizes. In this section, we outline the derivation of these methods, and formally show the derivation in Appendix B. We implement the adaptive time-step AB-methods because this family of methods are numerically stable with our scheme for the momentum equations (see Section 2.2). We then describe the constraints on the time-step imposed by the Courant–Fredrichs–Lewy (CFL) condition.

#### 3.3.1 Adaptive-Time Step Adams-Bashforth Methods

Our time integration scheme uses a family of time-step variable Adams-Bashforth integrative methods. This specific formulation of the AB methods allows for the model time step to be adjusted dynamically following the CFL conditions described in Section 3.3.2. Consider a tracer quantity $c$ that evolves according to

$$\frac{\partial c}{\partial t} = f(t, c(t)). \tag{94}$$

Here the function $f$ conceptually represents the entire model state, including the physical, biogeochemical, and non-conservative tendencies. We make a note here that the diffusive component of the time-integration step is calculated implicitly and not included in the ABIII integration step (see Equation (91)). We implement the third order Adams-Bashforth or ABIII method in this version of the model as the default option for time integration.

$$\text{ABIII:} \quad c(t_{n+1}) = c(t_n) + \frac{1}{6} \left( f^{n-2} \frac{\Delta t_{n+1}^2 (2\Delta t_{n+1} + 3\Delta t_n)}{\Delta t_{n-1}(\Delta t_n + \Delta t_{n-1})} - f^{n-1} \frac{\Delta t_{n+1}^2 (2\Delta t_{n+1} + 3\Delta t_n + 3\Delta t_{n-1})}{\Delta t_{n-1}\Delta t_n} \right.$$
$$\left. + f^n \frac{\Delta t_{n+1}(2\Delta t_{n+1}^2 + 6\Delta t_{n+1}\Delta t_n + 3\Delta t_{n+1}\Delta t_{n-1} + 6\Delta t_n^2 + 6\Delta t_n \Delta t_{n-1})}{(\Delta t_n + \Delta t_{n-1})\Delta t_n} \right). \tag{95}$$

The first two time-steps require the lower order methods. We implement a forward Euler for the first time step and a second-order AB scheme (defined below) for the second time step,

$$\text{ABII:} \quad c(t_{n+1}) = c(t_n) + \frac{\Delta t_{n+1}}{2\Delta t_n} \left( 2f(t_n, c(t_n))\Delta t_n + f(t_n, c(t_n))\Delta t_{n+1} - f(t_{n-1}, c(t_{n-1}))\Delta t_{n+1} \right). \tag{96}$$

Here the notation $\Delta t_n$ indicates the $n^{\text{th}}$ time step. Derivations for the adaptive time-stepping ABII and ABIII methods are given in Appendix B.

#### 3.3.2 CFL Conditions

MAMEBUS selects each model time step adaptively to ensure that time stepping is numerically stable. The time step is chosen to ensure that the CFL conditions for each of MAMEBUS's various advective and diffusive operators, described in preceding subsections, are satisfied.





The time step for advection of tracers is limited by the time scale associated with advective propagation across the width of a grid box ($\Delta_x$ or $\Delta_z$). These constraints can approximately be expressed as

$$\Delta t < \frac{\Delta_x}{|\overline{u}| + u_{\mathrm{igw}}}, \tag{97a}$$

$$\Delta t < \frac{\Delta_z}{|\overline{w}|}, \tag{97b}$$

(Durran, 2010). Here $u_{\mathrm{igw}}$ is the maximum horizontal propagation speed of internal gravity waves (Chelton et al., 1998),

$$c_{\mathrm{igw}} = \frac{1}{\pi} \int N \, \mathrm{d}z. \tag{98}$$

Particulate sinking in the NPZD model is also calculated explicitly and constrains the time step via a similar CFL criterion

$$\Delta t < \frac{(\Delta_z)}{|w_{\mathrm{sink}}|} \tag{99}$$

where $w_{\mathrm{sink}}$ is the sinking speed of the particles.

We apply additional constraints on the time step to ensure that diffusive operators are stable. The standard numerical stability criterion for a Laplacian diffusion operator is (Griffies, 2018)

$$\Delta t < \frac{1}{2} \frac{\Delta_s^2}{\kappa}, \tag{100}$$

where $\kappa$ is a diffusion coefficient and $\Delta_s$ is the spatial grid spacing. In the horizontal ($\Delta_s = \Delta_x$, the diffusion coefficients that determine the diffusive timestep are the eddy diffusion and isopycnal diffusion coefficients, when $\kappa = \kappa_{\mathrm{gm}}$ and $\kappa = \kappa_{\mathrm{iso}}$

respectively. [1] In the vertical ($\Delta_s = \Delta_z$), the diffusive time step is determined by the diapycnal diffusivity, $\kappa = \kappa_{\mathrm{dia}}$, and by the vertical component of the eddy and isopycnal diffusion operators, $\kappa = \kappa_{\mathrm{gm}} S_{\mathrm{int}}^2$ and $\kappa = \kappa_{\mathrm{iso}} S_{\mathrm{int}}^2$ respectively (see *e.g.* Ferrari et al., 2008).

### 3.4 Discrete Momentum Equations and Barotropic Pressure Correction

In this section, we describe the discretization of the momentum equations presented in Section 2.2, specifically in Equations
(11a), (11b), and (11c). To facilitate our discretization, we split the pressure $\phi$, into barotropic and baroclinic components,

$$\phi = \underbrace{\Pi}_{\text{barotropic}} + \underbrace{\frac{g}{\rho_0} \int_z^0 \rho \, \mathrm{d}z}_{\text{baroclinic}}. \tag{101}$$

The barotropic and baroclinic components correspond to the pressure at the surface and the hydrostatic pressure variation with depth, respectively.

The numerical time-integration is calculated in a series of steps which include an explicit calculation of the non-diffusive
time-step, an implicit calculation of the vertical diffusion, and a barotropic corrector step in order to ensure that the flow is

---

[1]Note that although $\kappa_{\mathrm{iso}}$ appears only in an advective operator in (83), this operator can be written as the divergence of a diffusion tensor (Griffies, 2018), and experience with MAMEBUS suggests that the more restrictive, diffusive formulation more accurately constrains the model time step.





non-divergent. The calculation of the explicit time-step is outlined in Section 3.3 and Appendix B. In order to be numerically consistent with the calculation of the streamfunction, the mean horizontal velocities $\overline{u}$ and $\overline{v}$ are stored on the western face of each grid cell. In Figure 2, these are labeled as the $u$ points. The time-step calculation is shown below, noting that the explicit components of the time-step, $\mathcal{E}\{\cdot\}$ are calculated following the ABIII methods outlined in Section 3.3, and the implicit diffusion $\mathcal{I}\{\cdot\}$ is calculated following Equation (91).

Given the mean momentum at time step $n$, $\overline{\mathbf{u}}^n$, we first perform the explicit component of the time-step to construct an estimate of $\overline{\mathbf{u}}^{n+1}$, denoted as $\overline{\mathbf{u}}^*$,

$$\overline{\mathbf{u}}^* = \overline{\mathbf{u}}^n + \mathcal{E}\left\{ -f\hat{z} \times \mathbf{u}^n - \frac{\partial\Pi}{\partial y}\hat{\mathbf{y}} - \frac{g}{\rho_0}\nabla\int_z \rho^n\, \mathrm{d}z \right\}. \tag{102}$$

Note that the zonal barotropic pressure gradient, $\partial_x\Pi\hat{\mathbf{y}}$, is excluded from this equation; this will be revisited in the final component of the time-step. The discretization of the horizontal pressure gradient terms in (102) described in Section 3.5.

We next compute the tendency due to vertical viscosity following equation (91), which we denote via the operator $\mathcal{I}$. We thereby construct a second estimate of the velocity at time step $n+1$, denoted as $\overline{\mathbf{u}}^{**}$,

$$\overline{\mathbf{u}}^{**} = \mathcal{I}\{\overline{\mathbf{u}}^*\}. \tag{103}$$

Finally, we apply a tendency due to the zonal barotropic pressure gradient, ensuring that mass is conserved in each vertical fluid column (Dauhajre and McWilliams, 2018),

$$\int_z \overline{u}^{n+1}\, \mathrm{d}z = \int_z \overline{u}^n\, \mathrm{d}z = 0, \tag{104}$$

as required by (11c). We formulate the barotropic pressure correction as

$$\overline{u}^{n+1} = \overline{u}^{**} - \Delta t\frac{\partial\Pi}{\partial x} \tag{105}$$

Substituting Equation (105) into (104), we obtain

$$\Delta t\frac{\partial\Pi}{\partial x}\hat{\mathbf{x}} = \frac{1}{|\eta_b(x)|}\int_z \overline{u}^{**}\, \mathrm{d}z. \tag{106}$$

This implies that the tendency in the mean zonal velocity due to the barotropic zonal pressure gradient must serve to bring the depth-integrated zonal velocity to zero, *i.e.*

$$\overline{u}^{n+1} = \overline{u}^{**} - \frac{1}{|\eta_b(x)|}\int_z \overline{\mathbf{u}}^{**}\, \mathrm{d}z. \tag{107}$$

The calculation of the vertical integral of $\overline{\mathbf{u}}^{**}$ is computed in the model using a Kahan sum (Kahan, 1965).

## 3.5 Horizontal Pressure- and Buoyancy-Gradient Calculations

Pressure gradient calculations in sigma coordinates have been long known to produce discretization errors (Arakawa and Suarez, 1983; Haney, 1991; Mellor et al., 1994, 1998). These errors arise from the misalignment of geopotential and sigma



coordinate surfaces. We follow Shchepetkin and McWilliams (2003) to calculate the horizontal pressure gradient force. For numerical consistency, we also calculate horizontal buoyancy gradients, required to evaluate the isopycnal slope (see §3.2), using the same algorithm.

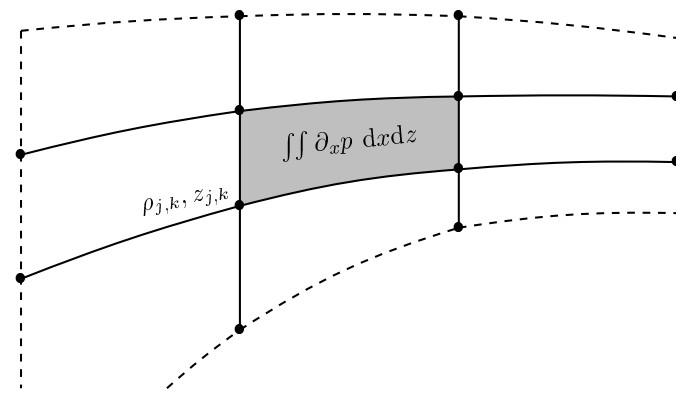

**Figure 3.** Stencil for the isopycnal slope and pressure gradient scheme given by Shchepetkin and McWilliams (2003). The points indicate the buoyancy (density) points. The solid lines are the reconstructed coordinate lines used in the horizontal calculation, and the shaded area shows the area integral of the horizontal buoyancy gradient.

### 3.5.1 Zonal Pressure Gradients

In this section, we outline the implementation of the baroclinic zonal pressure gradient calculation used in MAMEBUS following Shchepetkin and McWilliams (2003). The ultimate goal of the algorithm is to calculate the following baroclinic pressure gradient at a cell center (*c.f.* (102)),

$$-\frac{\partial \overline{\phi}}{\partial x}\bigg|_z^{\text{baroclinic}} = -\frac{1}{A}\frac{g}{\rho_0}\left(\iint_A \frac{\partial \rho}{\partial x}\bigg|_z \mathrm{d}x\mathrm{d}z\right), \tag{108}$$

where $\rho$, like the linear case, is the density anomaly, $\rho_0$ is a reference density, and $A$ is the area between four adjacent buoyancy grid points (shaded area in Figure 3. Shchepetkin and McWilliams (2003) calculate the second term by implementing a Lagrange polynomial reconstruction of the $z$ and $\rho$ fields. By Green's theorem, we write the integrated horizontal density gradient as

$$\iint_A \frac{\partial \rho}{\partial x}\bigg|_z \mathrm{d}x\mathrm{d}z = \oint \rho\mathrm{d}\mathbf{z}. \tag{109}$$

The two algorithms differ on the treatment of the vertical integration of pressure. The *Density Jacobian Algorithm* first interpolates the density field onto the sigma grid and calculates the values of $\rho$ and $z$ along the solid lines in Figure 3, then integrates to obtain the pressure. The *Sigma Coordinate Primitive Form Algorithm* first integrates the pressure and calculates the gradients using a vertical correction term. We opt for the second algorithm which results in an equation for the horizontal gradient which closely resembles a Equation (68a). Furthermore, this algorithm tends to be more stable in our model. The algorithm is calculated as follows:



First, we calculate all elementary differences in $\rho$ and $z$

$$(\Delta_x)\rho_{j+1/2,k} = \rho_{j+1,k} - \rho_{j,k}, \tag{110a}$$

$$(\Delta_z)\rho_{j,k+1/2} = \rho_{j,k+1} - \rho_{j,k}, \tag{110b}$$

$$(\Delta_x)z_{j+1/2,k} = z_{j+1,k} - z_{j,k}, \tag{110c}$$

$$(\Delta_z)z_{j,k+1/2} = z_{j,k+1} - z_{j,k}, \tag{110d}$$

where $z_{j,k}$ is the depth value at the cell centers, where the density tracer is located. Note that the edges of Figure 3 correspond to the cell centers in MAMEBUS, this requires some extrapolation at the boundaries so that the elementary differences are fully calculated throughout the domain. For all variables, we assume that the elementary differences at the boundary are zero.

We then calculate the hyperbolic differences of all variables. This step calculates an estimate of the derivatives following a cubic spline formalism outlined in more detail in Shchepetkin and McWilliams (2003). The derivatives are then given by,

$$h_x\rho_{j,k} = \frac{2(\Delta_x)\rho_{j+1/2,k}(\Delta_x)\rho_{j-1/2,k}}{(\Delta_x)\rho_{j+1/2,k} + (\Delta_x)\rho_{j-1/2,k}}, \tag{111a}$$

$$h_x z_{j,k} = \frac{2(\Delta_x)z_{j+1/2,k}(\Delta_x)z_{j-1/2,k}}{(\Delta_x)z_{j+1/2,k} + (\Delta_x)z_{j-1/2,k}}. \tag{111b}$$

The vertical hyperbolic differences for $\rho$ and $z$ are calculated similarly using Equations (110b) and (110d). Again, the hyperbolic differences on the boundaries are not defined using Equations (111a) and (111b), so we extrapolate the hyperbolic averages of density. For example, at the western edge of the domain we define

$$h_x\rho_{N,k} = \frac{3}{2}\left((\Delta_x)\rho_{N-1/2,k} - (\Delta_x)\rho_{N-3/2,k}\right) - \frac{1}{2}\left(h_x\rho_{N-1,k}\right) \tag{112}$$

Analogous extrapolation schemes are applied at all domain boundaries.

We the calculate the pressure field using the hydrostatic relationship. This is done via a vertical integration of the density reconstructed along the vertical lines in Figure 3. The pressure field is calculated from the surface down. The pressure is calculated in the surface grid cells as

$$\phi_{j,N} = \frac{g}{\rho_0}\left(\rho_{j,N} + \frac{1}{2}(\zeta_j - z_{j,N})\frac{\rho_{j,N} - \rho_{j,N-1}}{z_{j,N} - z_{j,N-1}}\right)(\zeta_j - z_{j,N}), \tag{113}$$

where $\zeta_j = 0$ from the rigid lid assumption in MAMEBUS. Then the pressure is calculated at successively deeper grid levels as,

$$\phi_{j,k} = \phi_{j,k+1} + \frac{g}{\rho_0}\left(\frac{\rho_{j,k+1} + \rho_{j,k}}{2}(z_{j,k+1} - z_{j,k}) - \frac{1}{10}\left\{(h_z\rho_{j,k+1} - h_z\rho_{j,k})\left[z_{j,k+1} - z_{j,k} - \frac{h_z z_{j,k+1} + h_z z_{j,k}}{12}\right]\right.\right.$$
$$\left.\left. -(h_z z_{j,k+1} - h_z z_{j,k})\left[\rho_{j,k+1} - \rho_{j,k} - \frac{h_z\rho_{j,k+1} + h_z\rho_{j,k}}{12}\right]\right\}\right). \tag{114}$$

We then correct for the iso-$\sigma$ pressure gradient introduced by the slope of the sigma coordinate grid, analogous to the continuous expression in Equation (68a). This step calculates the product of $\rho$ and the local slope of the $\sigma$-coordinate, and corrects





for the interpolation errors from the coordinate transformation. Following the notation used in Shchepetkin and McWilliams (2003),

$$FC_{j+1/2,k} = \frac{\rho_{j+1,k} + \rho_{j,k}}{2}(z_{j+1,k} - z_{j,k}) - \frac{1}{10}\left\{(h_x\rho_{j+1,k} - h_x\rho_{j,k})\left[z_{j+1,k} - z_{j,k} - \frac{h_x z_{j+1,} + h_x z_{j,k}}{12}\right]\right.$$
$$\left. -(h_x z_{j+1,k} - h_x z_{j,k})\left[\rho_{j+1,k} - \rho_{j,k} - \frac{h_x\rho_{j+1,k} + h_x\rho_{j,k}}{12}\right]\right\}. \tag{115}$$

Finally, we use Equations (114) and (115) to calculate the pressure gradients.

$$\left(\frac{\partial\phi}{\partial x}\bigg|_z\right)_{j+1/2,k} = \frac{1}{\Delta x}\left(\phi_{j+1,k} - \phi_{j,k} + \frac{g}{\rho_0}\cdot FC_{j+1/2,k}\right). \tag{116}$$

**Buoyancy Gradients**

The buoyancy gradient is calculated similarly to the pressure gradient. However, because we do not vertically integrate the buoyancy term, we opt to use the *Density Jacobian Algorithm* described in Shchepetkin and McWilliams (2003). The pressure

gradient algorithm described above integrates the pressure and then corrects for the pressure gradient in sigma coordinates. The density gradient algorithm described below calculates the line integral about the area enclosed by the $\phi$-points where the buoyancy gradient is located (see Figure 3). Therefore, we use the following form to calculate the buoyancy gradient,

$$\iint\limits_A \frac{\partial b}{\partial x}\mathrm{d}x\mathrm{d}z = \oint b\mathrm{d}\mathbf{z} = FX_{j+1,k+1/2} + FC_{j+1/2,k} - FX_{j,k+1/2} - FC_{j+1/2,k+1}, \tag{117}$$

where $FX_{j,k+1/2}$ are the value of the integral (Equation (117)) along the vertical sides, and $FC_{j+1/2,k}$ is the value of the

integral along the horizontal sides. This calculation follows a similar procedure as the pressure gradient.

First, we calculate the elementary differences, and the hyperbolic averages in $b$ and $z$, given by Equations (110a) through (111b). Then calculate the value of the integral along the upper and lower sides of the domain following,

$$FC_{j+1/2,k} = \frac{b_{j+1,k} + b_{j,k}}{2}(z_{j+1,k} - z_{j,k}) - \frac{1}{10}\left\{(h_x b_{j+1,k} - h_x b_{j,k})\left[z_{j+1,k} - z_{j,k} - \frac{h_x z_{j+1,} + h_x z_{j,k}}{12}\right]\right.$$
$$\left. -(h_x z_{j+1,k} - h_x z_{j,k})\left[b_{j+1,k} - b_{j,k} - \frac{h_x b_{j+1,k} + h_x b_{j,k}}{12}\right]\right\}. \tag{118}$$

Note that this formulation is the same as Equation (115), but with buoyancy instead of pressure. Then we calculate the value of the line integral along the vertical components of the cell,

$$FX_{j,k+1/2} = \frac{b_{j,k+1} + b_{j,k}}{2}(z_{j,k+1} - z_{j,k}) - \frac{1}{10}\left\{(h_z b_{j,k+1} - h_z b_{j,k})\left[z_{j,k+1} - z_{j,k} - \frac{h_z z_{j,k+1} + h_z z_{j,k}}{12}\right]\right.$$
$$\left. -(h_z z_{j,k+1} - h_z z_{j,k})\left[b_{j,k+1} - b_{j,k} - \frac{h_z b_{j,k+1} + h_z b_{j,k}}{12}\right]\right\}. \tag{119}$$

Shchepetkin and McWilliams (2003) write Equation 68a as,

$$\frac{\partial b}{\partial x}\bigg|_z = \mathcal{J}(b,z) = \frac{\partial b}{\partial x}\bigg|_\sigma \frac{\partial z}{\partial\sigma} - \frac{\partial b}{\partial\sigma}\frac{\partial z}{\partial x}\bigg|_\sigma, \tag{120}$$





This allows us to numerically integrate the buoyancy gradient in the cell as,

$$\oint b\mathrm{d}\mathbf{z} = \left( A\frac{\partial b}{\partial x} \right)_{j+1/2,k+1/2} = FX_{j+1,k+1/2} + FC_{j+1/2,k} - FX_{j,k+1/2} - FC_{j+1/2,k+1} \tag{121}$$

where, $A$, again is the area of the cell. At the surface, the boundary condition is given that $FC_{j+1/2,N+1} \equiv 0$.

Finally, in order to calculate the horizontal buoyancy gradient, we divide by the area. Since, the area of each cell is defined

by the cell-centered, $\phi$-points, we implement Gauss' Area Formula,

$$A_{j+1/2,k+1/2} = \frac{1}{2} |x_{j,k+1}z_{j,k} + x_{j,k}z_{j+1,k} + x_{j+1,k}z_{j+1,k+1} + x_{j+1,k+1}z_{j,k+1}$$

$$-x_{j,k}z_{j,k+1} - x_{j+1,k}z_{j,k} - x_{j+1,k+1}z_{j+1,k} - x_{j,k+1}z_{j+1,k+1}| \tag{122}$$

### 3.5.2 Meridional Pressure Gradients

The alongshore pressure gradient in Equation (11b), denoted by $\left[\phi\right]_0^{L_y}/L_y$, and is determined by along-shore gradients in

the surface pressure and buoyancy/density that are imposed as model input parameters. We integrate the profiles of pressure

following the hydrostatic relationship. We define $\rho^N$ and $\rho^S$ as the densities at the northern and southern ends of the domain,

and $\Pi^y = \Pi^{N,S}$ as the surface pressures at northern and southern ends of the domain. Then the pressure is given by,

$$\frac{1}{L_y}\left[\phi\right]_0^{L_y} = \frac{\partial \Pi}{\partial y} - \frac{g}{\rho_0}\int_z^0 \frac{\partial \rho}{\partial y}\,\mathrm{d}z,$$

$$= \frac{\Pi^N - \Pi^S}{\rho_0 L_y} + \frac{g}{\rho_0}\int_z^0 \frac{\rho^N - \rho^S}{L_y}\,\mathrm{d}z. \tag{123}$$

Here the along shore variations in sea surface pressure and density are both model inputs. We discretize the meridional pressure

gradient as,

$$\frac{1}{L_y}\left[\phi\right]_0^{L_y}\bigg|_{j,k} = \frac{1}{L_y}\left[ \left(\Pi_{j,N}^N - \Pi_{j,N}^S\right) + \frac{g}{\rho_0}\sum_{k=k'}^{N-1}\left( \frac{\rho_{j,k'+1}^N + \rho_{j,k'}^N}{2} - \frac{\rho_{j,k'+1}^S + \rho_{j,k'}^S}{2} \right)\left(z_{j,k'+1} - z_{j,k'}\right) \right]. \tag{124}$$





## 4  Implementation Details

In this section, we outline the details for implementation in MAMEBUS. The model code is written in the C programming
language. The model expects various user inputs that include initial conditions, along with user-defined model calculation
details in Table 4 that include, but are not limited to, the momentum calculation scheme and the time-stepping scheme. The
5   MAMEBUS distribution includes sample Matlab codes that package these user inputs.

MAMEBUS has three active physical variables: the zonal and meridional momenta, and the temperature (buoyancy). The
current implementation of the biogeochemical model has four active variables: nitrate (N), phytoplankton (P), zooplankton (Z),
and detritus (D). A variable number of additional passive tracers may also be included.

### 4.1  Expected User-inputs, and Options Available

**Table 3.** Input parameters expected by the MAMEBUS model code. All parameters listed in this table are chosen by the user. The sample
values listed in this table are those used in the reference experiments described in Section 5.

| Description | Value | Units |
|---|---|---|
| Number of horizontal grid points | 64 | |
| Number of vertical grid points | 64 | |
| Computational domain width | 400 | km |
| Computational domain height | 3000 | m |
| Depth of the shelf | 50 | m |
| Location of the continental slope in the domain from the eastern boundary | 50 | km |
| Topographic slope | 9.8e-3 | |
| Depth of surface mixed layer | 40 | m |
| Depth of bottom boundary layer | 40 | m |
| Drag coefficient in the bottom boundary layer | 1e−3 | m/s |
| Reference density | 1000 | kg/m$^3$ |
| Coriolis parameter | 1e−4 | 1/s |
| Surface grid stretching parameter | 9 | |
| Bottom grid stretching parameter | 4 | |
| Depth below the surface over over which the vertical coordinate the coordinate is approximately aligned with geopotentials | 300 | m |
| The fraction of the maximum time-step taken for each $\Delta t$ to ensure the CFL condition is met | 0.75 | |
| The end time for integration | 30 | years |
| Output frequency of model data | 1 | day |





MAMEBUS expects a list of parameters given in Table 3, that control the physical components of the model, the model run details, and the grid setup. Other identifiers included in this model are given in Table 4 which determine which internal schemes the model uses for each specific run. Furthermore, MAMEBUS expects a set of input parameters from physical tracers, forcing, diffusivity, and restoring, along with initial profiles of biogeochemical tracers that are listed in Table 5.

5    For the solutions shown in Section 5, the following intial conditions are detailed in Section 5.1.

**Table 4.** MAMEBUS numerical scheme options and descriptors.

| Parameter | Identifier | Value | Scheme description |
|---|---|---|---|
| modelType | BGC_NONE | 0 | Physics only, no biogeochemistry |
| | BGC_NPZD | 1 | Nutrient-Phytoplankton-Zooplankton-Detritus (NPZD) Model, described in Section 2.4 |
| timeSteppingScheme | TIMESTEPPING_AB1 | 0 | First-order Adams Bashforth variable timestepping |
| | TIMESTEPPING_AB2 | 1 | Second-order Adams Bashforth variable timestepping |
| | TIMESTEPPING_AB3 | 2 | Third-order Adams Bashforth variable timestepping |

**Table 5.** A table outlining the initial profiles that MAMEBUS expects during initializaiton. To visualize the grid locations, see Figure 2. Each initial profile is included in all modelTypes unless otherwise stated. Note that Nx is the number of zonal domain points and Nz is the number of vertical domain points given in Table 3.

| Initial Profile | Parameter | Grid Location | Size | Descriptions |
|---|---|---|---|---|
| Zonal Momentum | $u(x,z)$ | $u$-points | Nx+1 $\times$ Nz | All modelTypes |
| Meridional Momentum | $v(x,z)$ | $v$-points | Nx+1 $\times$ Nz | All modelTypes |
| Temperature | $T$ | $\phi$-points | Nx $\times$ Nz | All modelTypes |
| Nitrate | $N$ | $\phi$-points | Nx $\times$ Nz | NPZD Model |
| Phytoplankton | $P$ | $\phi$-points | Nx $\times$ Nz | NPZD model |
| Zooplankton | $Z$ | $\phi$-points | Nx $\times$ Nz | NPZD model |
| Detritus | $D$ | $\phi$-points | Nx $\times$ Nz | NPZD model |
| Buoyancy Diffusivity | $\kappa_{\mathrm{gm}}$ | $\psi$-points | Nx+1 $\times$ Nz+1 | See Equation (131) |
| Isopycnal Diffusivity | $\kappa_{\mathrm{iso}}$ | $\psi$-points | Nx+1 $\times$ Nz+1 | See Section 5.2 |
| Topography | $h_b(x)$ | $w,\psi$-points | Nx+1 $\times$ 1 | See Equation (126) |
| Wind Stress | $\tau(x,t)$ | $\psi$-points | Nx+1 $\times$ 1 | See Equation (125) |

### 4.2 Model Run Details

The main function of the mamebus.c file has five major components and steps:

1. Calculate the time tendency of each tracer. The time step is calculated using the tderiv function detailed in Figure 4. The explicit tendencies are calculated following Section 2.





2. Add implicit vertical diffusion and remineralization, Equation (91).

3. Apply zonal barotropic pressure gradient correction if the momentumScheme is MOMENTUM_TTW (Section 3.4 )

4. Enforce zero tendency where relaxation time is zero (Section 2.5.1).

5. Write model state (Section 4.3).

## 4.3 Model Data

All of the model input and output are saved in binary files. Depending on the "monitorFreq" or the frequency of output, the model will interpolate the between timesteps, if necessary, calculate the correct model state, and write the data to file. The following list contains all files that are written to file during the time integration step. For each model, there is an option to include an arbitrary number of passive tracers, however these are the standard list of tracers that are included in the indicated modelTypes.

- Residual Streamfunction, $\psi^\dagger$, (all modelTypes)

- Mean Streamfunction, $\overline{\psi}$, (all modelTypes)

- Eddy Streamfunciton, $\psi^\star$, (all modelTypes)

- Temperature field, (all modelTypes)

- Nitrate, (NPZD model)

- Phytoplankton (NPZD model)

- Zooplankton (NPZD model)

- Detritus (NPZD model)





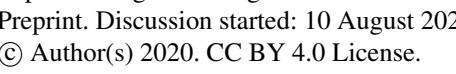

**Figure 4.** The call tree from the main function of mamebus.c





# 5 Reference Solution and Model Validation

In this section we present reference solutions for MAMEBUS. Below we discuss the choice of parameters, the non-conservative forcing, and profiles of restoring. We focus predominantly on the output of a single run, and plan in the future to run parameter sweeps to better understand the response of the ecosystem dynamics to the physical forcing.

5 ## 5.1 Model Geometry, Initial Conditions and Forcing

The model is configured to represent an idealized California Current System (CCS). While the model can be formulated to represent a general EBUS, we use the California Current System as a test case because this allows comparison of our results with measurements from California Cooperative Oceanic Fisheries Investigations (McClatchie, 2016). Note that we exclude salinity as a physical tracer; while it may be important in determining the structure of the California undercurrent (Connolly
10 et al., 2014), we find that the main features of stratification can be well described by temperature.



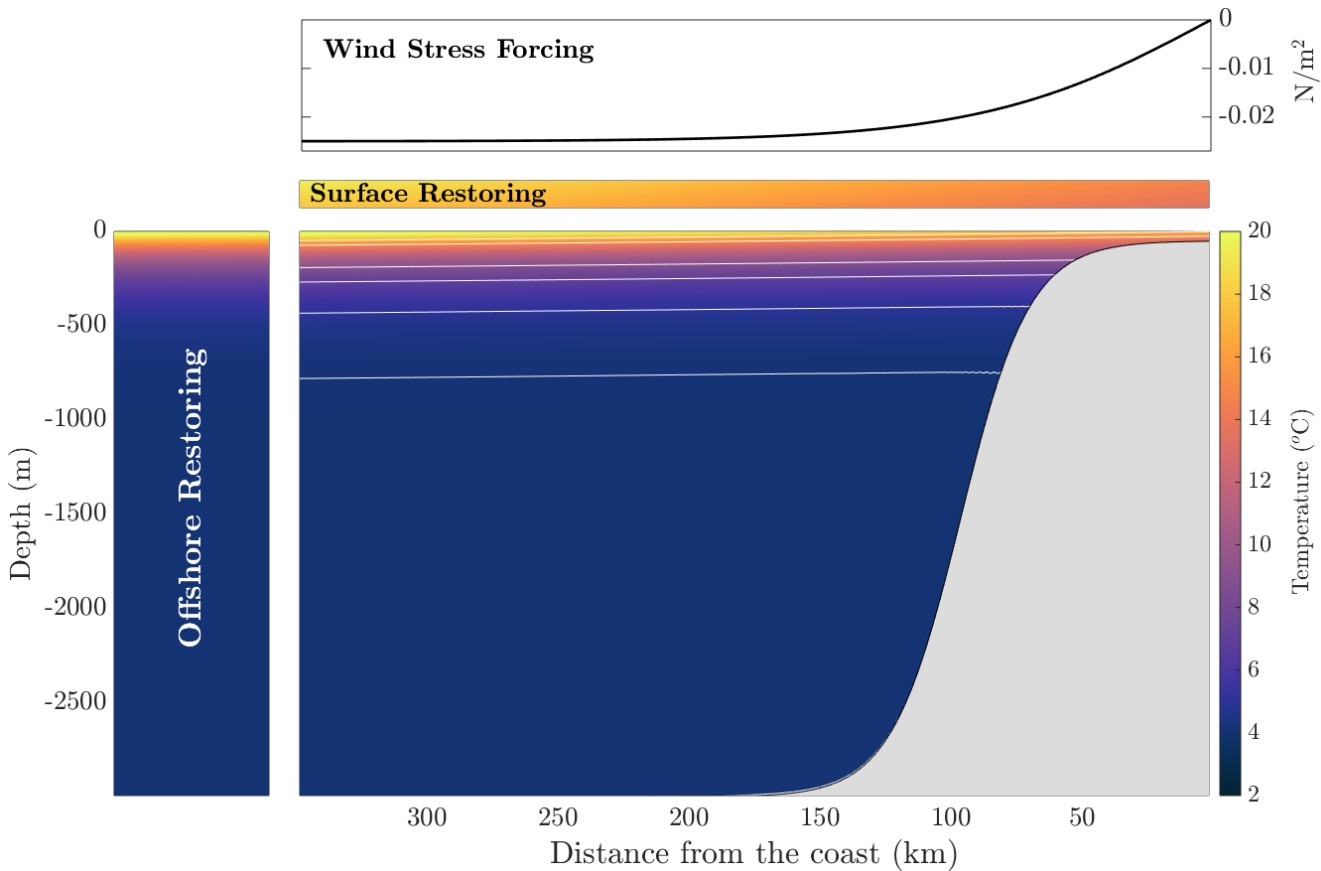

**Figure 5.** Initial temperature profile with a profile of offshore restoring which is modeled as a sponge layer on the western side of the boundary, and at the surface, there is a surface restoring to an atmospheric profile, idealized to a profile of temperature from the California Cooperative Oceanic Fisheries Investigations (CalCOFI). The northward wind stress is shown at the top of the figure. The white lines in the temperature field are a few lines of constant initial temperature.

A list of input fields that MAMEBUS expects is given in Table 5, with a subset illustrated in Figure 5. The solutions shown in Section 5 use he following choices for these input fields. The wind stress profile is given by

$$\tau(x) = \tau_0 \tanh\left(\lambda_\tau \frac{L_x - x}{L_x}\right), \tag{125}$$

where $L_x$ is the width of the computational profile given in Table 3, and $\lambda_\tau = 4$ is a tuning parameter that controls the horizontal width of the wind stress drop off, or wind stress curl. This profile is tuned to give an idealization of a wind stress curl profile from maps shown in Castelao and Luo (2018).

The topography for the reference solutions is ,

$$\eta_b(x) = H - \frac{1}{2}(H - H_s)\tanh\left(\frac{x - x_t}{L_t}\right), \tag{126}$$





where $H$ is depth of the computational domain, $H_s$ is the slope depth, $x_t$ is the location of the continential slope in the computational domain, and $L_t$ is the width of the continental slope given from the topographic slope parameter. All parameters are given in Table 3. The topography is tuned to represent an idealized profile of bathymetry ETOPO5 (Eto, 1988) taken from the geographic coordinates given from Line 80 in the CalCOFI data (McClatchie, 2016).

The initial conditions for the tracers in the model are the intial temperature profile, including timescales and inputs for restoring, and initial conditions for the NPZD model, which are tuned to give an approximate concentration of 30 mmol/m$^3$ in the deep ocean. The biogeochemical tracers are not restored in this set of reference solutions. The initial profile of temperature is shown in Figure 5 and given by,

$$T_{\text{init}}(x,z) = T_{\min} + (T_{\max} - T_{\min}) \frac{\exp\left(\frac{z}{H^*} + 1\right) - \exp\left(-\frac{H}{H^*} + 1\right)}{\exp(1) - \exp\left(-\frac{H}{H^*} + 1\right)}, \tag{127}$$

where the minimum and maximum temperatures in the domain are $T_{\min} = 4°C$, $T_{\max} = T_{\max}^s - (T_{\max}^s - T_{\min}^s)x/L_x$. The maximum and minimum surface temperatures are $T_{\max}^s = 22°C$ and $T_{\min}^s = 18°C$, respectively. $H^*$ is a decay scale for the temperature from the surface. This profile is tuned so that the temperature profile on the western side of the domain approximately matches the profile of temperature from CalCOFI (McClatchie, 2016) in Figure 7. We initialize the temperature field with a small tilt in the iso-surfaces to speed up the spin-up process. This same initial condition is used as the reference for temperature

restoring. The timescale for restoring is given by

$$R_T^{\text{west}}(x,z) = \left(\frac{1}{R_T^{\max}} \frac{L_r - x}{L_r}\right)^{-1}, \qquad x < L_r \tag{128}$$

where $L_r = 50$km is the width of the sponge layer on the western side of the domain, and $R_T^{\max} = 30$ days is the fastest relaxation timescale for temperature. In the surface grid boxes, the restoring timescale is given by,

$$R_T^{\text{surf}}(x) = 1 \text{ day}, \tag{129}$$

which is consistent with the formulation of Haney (1971) for surface grid box thicknesses of approximately $1\,\text{m}$.

    The initial conditions for NPZD tracers a a constant concentration of nitrate, $N_{\max} = 30$ mmol/m$^3$, phytoplankton $P_{\max} = 0.02$ mmol/m$^3$, zooplankton, $Z_{\max} = 0.01$ mmol/m$^3$, and an initial profile of detritus of zero. This choice allows for the internal ecosystem dynamics to control the biogeochemical solutions. Finally, the cell sizes we choose for the phytoplankton cell is, $\ell_p = 1\mu$m. The zooplankton cell is optimized to give the optimal predator-prey length scale between the phytoplankton and

zooplankton interactions, ie,

$$\ell_z = \exp\left(\frac{1}{0.56} \log\left(\frac{\ell_p}{0.65}\right)\right). \tag{130}$$





## 5.2 Isopycnal, buoyancy, and diapycnal mixing

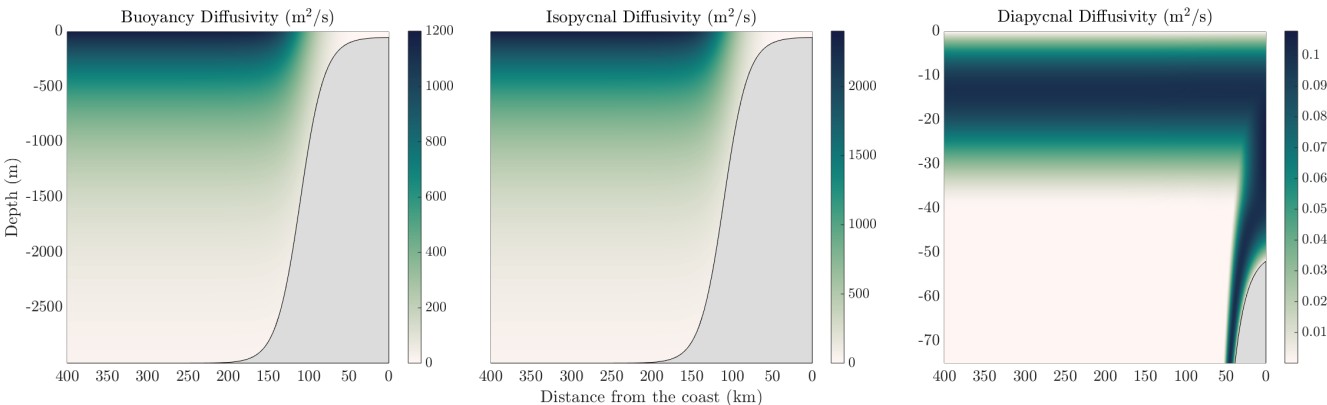

**Figure 6.** Inputs of buoyancy diffusivity (left), isopycnal diffusivity (center), and diapycnal diffusivity (right) used in the reference solution to MAMEBUS shown in Section 5. Note that the isopyncal and diapycnal diffusivities are shown over the entire domain, and the diapycnal diffusivities are shown over the upper 75m of the domain to highlight the boundary layer mixing and the mixing in the eastern side of the domain on the shelf where the boundary layers merge.

The unresolved mesoscale and microscale mixing in the tracer evolution Equation (9) are detailed in Sections 2.3.1 and 2.3.2, respectively. The diapycnal diffusivities are independent of wind-stress, and are determined by user-input mixed layer depth and maximum magnitudes. The isopycnal and buoyancy diffusivities are time-invariant fields whose spatial structure is prescribed by the user.

In our model reference configuration, the eddy and buoyancy diffusivities are functions of the baroclinic radius of deformation – the preferential length scale at which baroclinic instability occurs, and closest to the fastest growing mode in the Eady model (Eady, 1949). In MAMEBUS, these diffusivities also exponentially decreases with depth. There are choices for more sophisticated parameterzations of eddy transfer acros continental slopes (Wang and Stewart, 2018, 2020), but in this current version of the model, we opt for a simpler description. For example, the buoyancy diffusivity coefficient is defined as:

$$\kappa_{\mathrm{gm}} = \frac{\kappa_{\mathrm{gm}}^0 R_d}{H_{\max}} \exp\left(\lambda \frac{z}{\eta_b}\right),$$ (131)

where $\lambda < 1$ is a tuning coefficient that allows for adjustment of the depth of the exponential profile of diffusivity, $H_{\max}$ is the maximum depth of the topography offshore, and $\eta_b$ is the depth of the topography. For all solutions shown in this section $\lambda = 0.25$ Note that for this formulation, we assume that $z < 0$. The maximum buoyancy diffusivity is $\kappa_{\mathrm{gm}}^0 = 1200$ m²/s. Furthermore, $\kappa_{\mathrm{iso}} = 2\kappa_{\mathrm{gm}}$, following Smith and Marshall (2009) and Abernathey and Marshall (2013). The isopycnal and buoyancy diffusivity profiles are shown in the left and center panels of Figure 6, respectively.

The diapycnal diffusivities shown in the right panel of Figure 6, with structure function described in Equation (17), are set so that the maximum diffusivity in the mixed layers are given by, $\kappa_{\mathrm{sml}}^0 = \kappa_{\mathrm{bbl}}^0 = 0.1$ m²/s, otherwise the ambient diffusivity in



the interior is given by, $\kappa_{\mathrm{bg}} = $ 1e-5 m$^2$/s. In the case where the mixed layers join at the eastern edge of the domain, the profiles of diffusivity are simply added.

## 5.3 Model Validation

We run the reference solutions of MAMEBUS for 25 model years, with initial conditions and physical forcing described in Section 5.1. We validate the model against observations of temperature, nitrate, and chlorophyll-a concentration in the euphotic zone, based on observations from the CalCOFI program (McClatchie, 2016). For this comparison, we interpolate a typical CalCOFI section (Line 80) to a sigma coordinate grid with realistic topography from the ETOPO database (Eto, 1988). We chose to validate our model with a single transect of from CalCOFI instead of several transects along the same line because averaging over time smooths over the deep chlorophyll maximum.

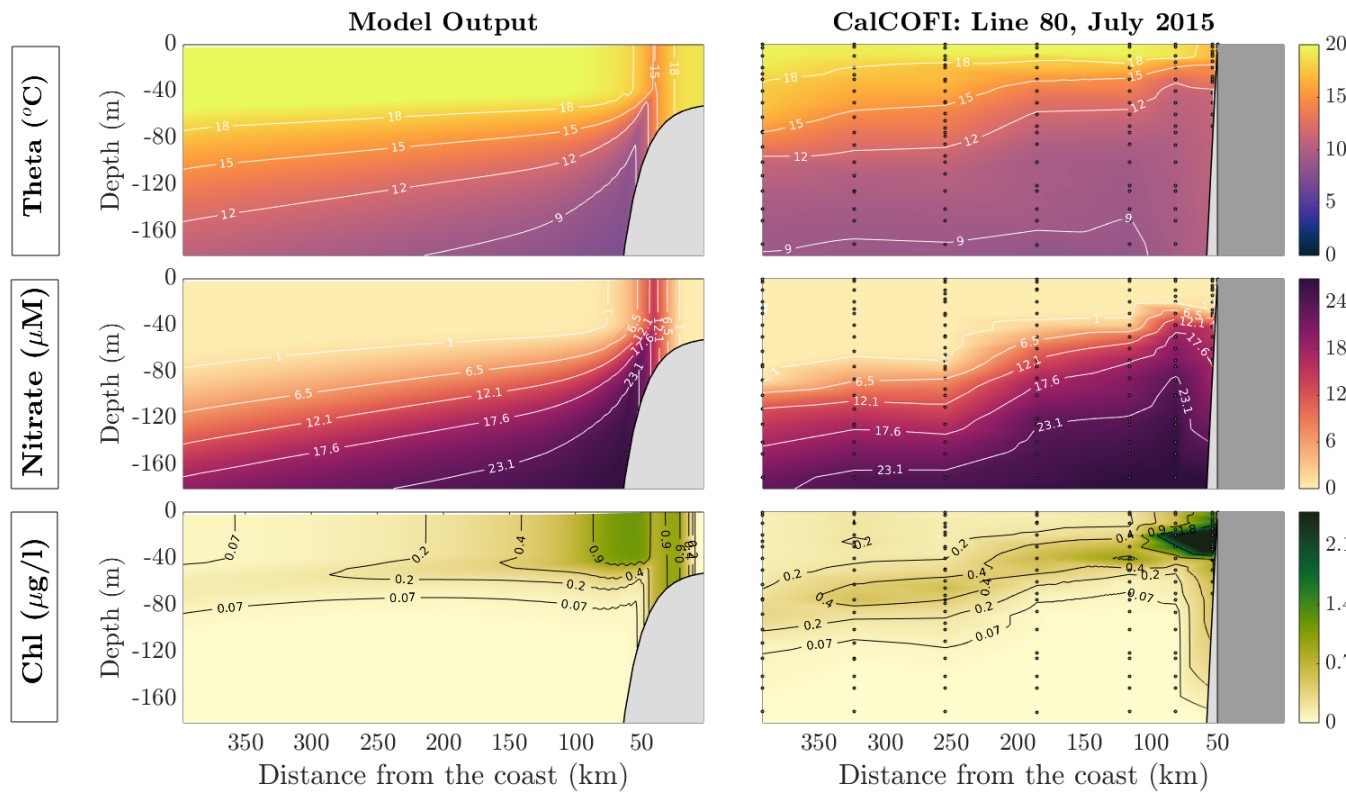

**Figure 7.** Model validation against *in situ* California Cooperative Oceanic Fisheries Investigations (CalCOFI) data taken along Line 80 (Point Conception) during July of 2015. The column on the left shows output from the model under constant wind forcing and is averaged over the last five model years. The column on the right are values taken from CalCOFI and interpolated onto a sigma coordinate grid to allow for direct comparison. The dots on the figures are locations where the data is sampled. This figure shows the comparison between potential temperature, $\theta$ (top), nitrate (middle), and chlorophyll concentration (bottom).





The model temperature is generally in good agreement with observations for the upper ocean, reproducing sloping isotherms towards the coast, and realistic surface values. We observe a cold bias near the coast, which could be a result of the constant wind-stress curl forcing over the domain, inducing upwelling that is too strong in the model. A cold bias observed in the surface just outside the shelf, and a warm bias offshore, are likely caused by the prescription of a constant mixed layer depth, which

may be too deep in the model for this particular section and time of the year.

As shown by the middle row of Figure 7 model nitrate agrees reasonably well with observations in the upper layers, although biases remain, in particular in deeper layers. This may be caused by several factors, including biases in the cross-shore and vertical circulation, and in the cycling of inorganic nutrients and organic matter. For example, remineralization processes are simplified in the model, which does not include dissolved organic matter, and represents export by a single particle size class

with a constant sinking speed that was not explicitly tuned to match nutrients.

The bottom row of Figure 7 shows that the model captures the main features of the observed chlorophyll distribution (here calculated based on a fixed chlorophyll to phytoplankton nitrogen ratio of 4:5 mg, chl/m$^3$:mmol N/m$^3$ following Furuya (1990)). High surface concentrations are reproduced near the shelf, with values decreasing further offshore. A deep chlorophyll maximum develops in the lower euphotic zone, at depths between 40 and 80 m, progressively deepening from the coastal to

the oligotrophic region offshore. while these patterns are fairly realistic, we note that the very high chlorophyll concentrations observed near the shelf are missing from the model. This underestimate may be caused by the over-simplification of the ecosystem structure in the NPZD model, which only includes a single phytoplankton group, while multiple groups are likely required for a more correct representation of enhanced coastal phytoplankton biomass (Van Oostende et al., 2018).

In order to compare physical solutions, we also include solutions which show the residual streamfunction, including the

mean and eddy components in Figure 8. The mean streamfunction is calculated via the momentum equations given in Section 2.2, whereas the eddy streamfunction is described in Section 2.3. The positive values indicate clockwise circulation, which, in this case, is indicative of eddy restratification opposing the mean upwelling branch (Colas et al., 2013). The negative values indicate counterclockwise circulation. Figure 8 shows that residual upwelling of waters onto the continental shelf via the bottom boundary layer, as interior transport onto the shelf is compensated by eddies. In the deep ocean ($> 500$ m there is a

relatively strong residual overturning circulation that is likely associated with bottom intensification of the diapycnal mixing coefficient (McDougall and Ferrari, 2017, *e.g.*).

## 5.4  Model Verification (Resolution Tests)

In this section, we describe the changes in solutions due to model resolution. We chose four different resolutions, which include $Nx = Nz = 32, 64, 96, 128$, and explore the results. Figure 9 shows the solutions of MAMEBUS after 30 model years. Each

panel is averaged over the final 10 model years, and has the same setup and forcing as described in Sections 4 and 5. Each row shows a different tracer, and each column shows the solution of each tracer at a different resolution. With increasing resolution, the depth of the nutracline shoals toward the surface, resulting in a higher concentration of phytoplankton at the surface offshore.



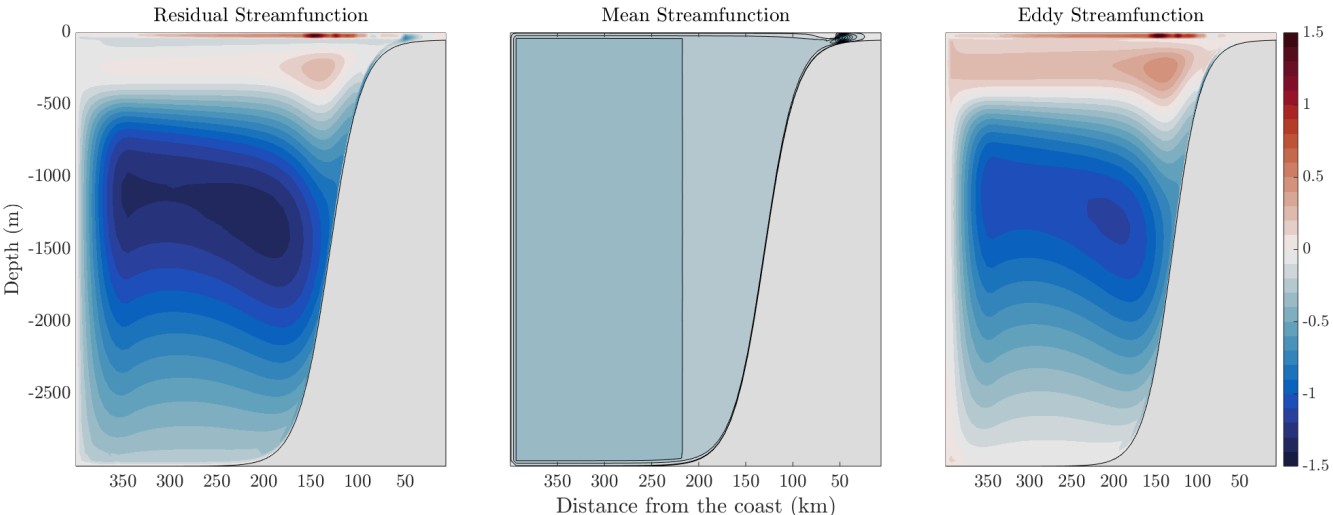

**Figure 8.** Streamfunctions calculated by MAMEBUS. This figure shows the residual streamfunction (left), $\psi^\dagger = \overline{\psi} + \psi^\star$, the mean streamfunction (center) as calculated in Section 2.2, and the eddy streamfunction (right) as described in Section 2.3. Note that positive values indicate clockwise circulation, whereas negative values indicate counter-clockwise circulation.

## 6  Discussion and Future Work

In this paper, we described the formulation, implementation, and main features of MAMEBUS, an idealized, meridionally-averaged model of eastern boundary upwelling systems. The model is based on solution of a general evolution equations for materially conserved tracers (Section 2) and the fluid momentum equations under the time-dependent turbulent thermal wind
(T3W) approximation (Dauhajre and McWilliams, 2018). It includes parameterizations of mesoscale eddy transfer and surface and bottom boundary layer mixing (Section 2.3), and a simple ecosystem formulation (Section 2.4). We further detailed the algorithms and discretizations implemented in the model (Section 3), and discussed reference model inputs and solutions (Section 4). Finally we performeda preliminary validation based on observations from the California Current system, and we discussed the sensitivity of the model to horizontal and vertical resolution (Section 5).

MAMEBUS represents a simple, physically-consistent tool in which to test and tune a variety of physical parameterizations and ecosystem model formulations. The ultimate goals of this research include exploration of physical-biogeochemical interactions in EBUS, mechanistic understanding of the factors that control cross-shore gradients in biogeochemical and ecological properties, and investigation of the processes that drive differences between distinct EBUS.

Because of the 2D framework, we acknowledge shortcomings to the model formulation, including physical aspects like
intensification of upwelling around topographic features, for example resulting from variations in the wind-stress curl (Castelao and Luo, 2018) or fine-scale ocean dynamics. Furthermore, while we parameterize the effect of mesoscale eddies on circulation, we do not account for submesoscale eddies on the shelf, which could play an important role in tracer transport (Dauhajre and

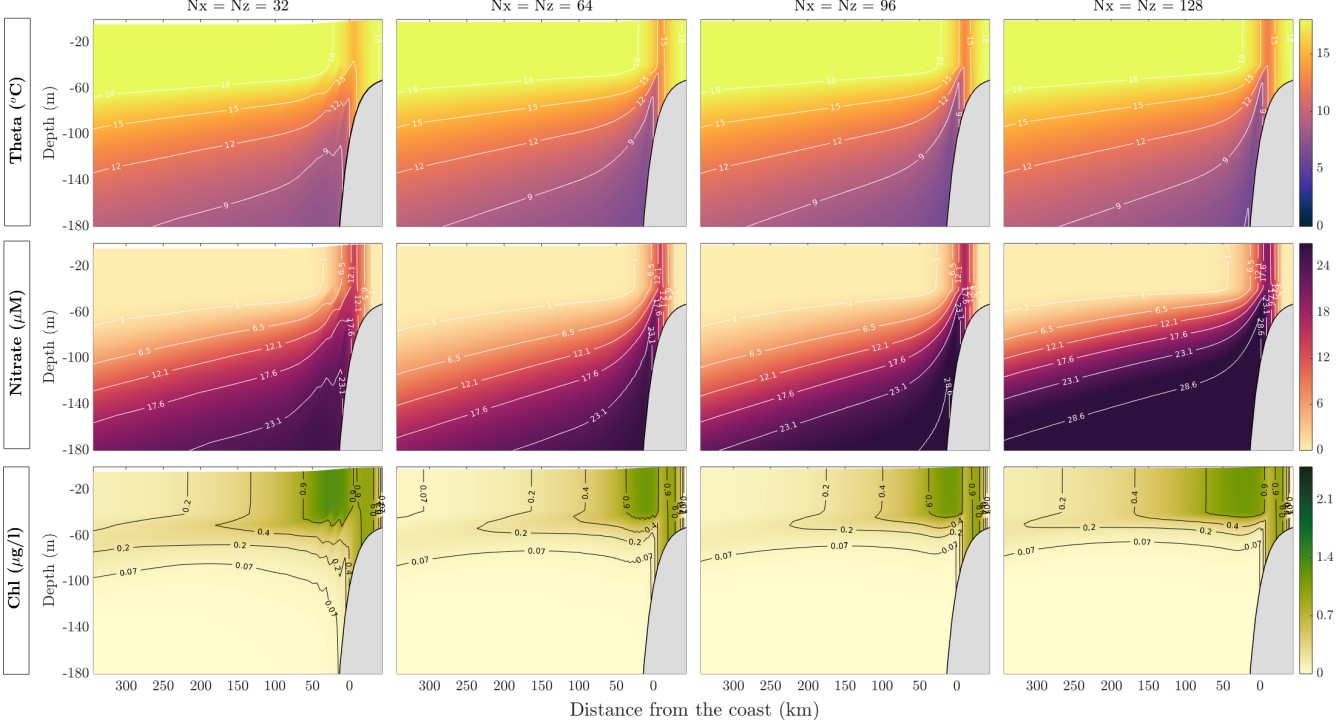

**Figure 9.** This figure shows the model output of temperature, nitrate, and chl with varying resolution. The model was run for 30 years, and solutions shown are averaged over the final 30 years of the model run.

McWilliams, 2018). We also do not explicitly represent breaking internal waves and tides on the shelf, which may play an important role in dissipating energy and mixing tracers when the water column is shallow (Lamb, 2014).

In future studies, we plan to use MAMEBUS to explore the effect of physical drivers such as wind stress, bathymetry, stratification, and eddies, in controlling the zonal distribution of phytoplankton and food web processes, as informed by a size-
5    structured ecosystem model. Furthermore, we plan to expand upon the physical framework in this paper by expanding eddy parameterizations to include the effect of submesoscale eddies on the shelf, where the mesoscale eddy activity is inhibited. An aspect of MAMEBUS that requires further investigation is the effect of meridional pressure gradients, which we neglected in our reference solutions in Section 5. In reality, the presence of along-shore pressure gradients may supports interior across-shore transport away from the surface and bottom boundary layers, with the potential to reshape the coastal ecosystem.

10    With its limited computational cost, MAMEBUS can be used to investigate a wide parameter space in EBUSs, and determine their sensitivity to a range of perturbations in major physical forcings, from changes in wind-stress to increasing buoyancy forcing associated with climate change (Rykaczewski and Dunne, 2010; Sarmiento et al., 1998). Furthermore, by allowing coupling to a variety of biogeochemical and ecosystem models, MAMEBUS can be used to inform comprehensive regional models (Shchepetkin and McWilliams, 2005), for which computational costs preclude exhaustive sensitivity studies.





*Code availability.* The DOI for the mamebus code is: 10.5281/zenodo.3866652

This package includes the mamebus.c code along with example setup and processing functions that are used in Matlab.

## A  Decomposing Mesoscale Eddy Advective/Diffusive Fluxes

In this Appendix we discuss the partitioning of the mesoscale eddy tracer flux into components due to advection and isopycnal

diffusion, used in Section 2.1 to derive MAMEBUS's central tracer evolution equation, (9). We show that the eddy tracer flux, $\overline{\mathbf{u}'c'}$, can be arbitrarily decomposed into components directed along mean buoyancy surfaces and along mean tracer surfaces. These components will later be associated with eddy advection and isopycnal stirring, respectively.

The effect of mesoscale eddies on the averaged tracer concentrations is given by the convergence of the eddy tracer flux (8),

$$\left.\frac{\partial \overline{c}}{\partial t}\right|_{\text{eddies}} = -\nabla \cdot \left(\overline{\mathbf{u}'c'}\right), \tag{A1}$$

and appears on the right-hand side of (6). Being quasi-adiabatic flows, mesoscale eddies serve to stir material tracers along isopycnal surfaces; this corresponds to an eddy tracer flux directed along buoyancy surfaces (Redi, 1982). Eddies also induce a "bolus" advective transfer of tracers, a generalized "Stokes drift" that corresponds to an eddy tracer flux directed along mean tracer iso-surfaces (Gent and McWilliams, 1990). Both of these effects are routinely parameterized in general circulation models (Griffies, 1998, 2018). To partition the eddy tracer flux between isopycnal stirring and bolus advection, we therefore

pose a decomposition of $\overline{\mathbf{u}'c'}$ into components directed along mean isopycnals and along mean tracer surfaces, respectively,

$$\overline{\mathbf{u}'c'} = \alpha_c \hat{\boldsymbol{\tau}}_c + \alpha_b \hat{\boldsymbol{\tau}}_b. \tag{A2}$$

Here $\hat{\boldsymbol{\tau}}_c$ and $\hat{\boldsymbol{\tau}}_b$ are unit vectors that point along mean $\overline{c}$ surfaces and along mean $\overline{b}$ surfaces, respectively:

$$\hat{\boldsymbol{\tau}}_c = \hat{\mathbf{y}} \times \frac{\nabla \overline{c}}{\|\nabla \overline{c}\|}, \qquad \hat{\boldsymbol{\tau}}_b = \hat{\mathbf{y}} \times \frac{\nabla \overline{b}}{\|\nabla \overline{b}\|}. \tag{A3}$$

Note that the $x$-components of $\hat{\boldsymbol{\tau}}_b$ and $\hat{\boldsymbol{\tau}}_c$ are positive provided that $\overline{b}$ and $\overline{c}$ increase monotonically upward. By taking the

vector cross products $\hat{\boldsymbol{\tau}}_c \times$ (A2) and $\hat{\boldsymbol{\tau}}_b \times$ (A2), we can solve for the vector lengths $\alpha_c$ and $\alpha_b$,

$$\alpha_c = \frac{\overline{\mathbf{u}'c'} \times \hat{\boldsymbol{\tau}}_b}{\hat{\boldsymbol{\tau}}_c \times \hat{\boldsymbol{\tau}}_b}, \qquad \alpha_b = \frac{\overline{\mathbf{u}'c'} \times \hat{\boldsymbol{\tau}}_c}{\hat{\boldsymbol{\tau}}_b \times \hat{\boldsymbol{\tau}}_c}. \tag{A4}$$

Then, using Equations (A2)–(A4), we write Equation (A1) as

$$\left.\frac{\partial \overline{c}}{\partial t}\right|_{\text{eddies}} = -\left(\nabla \times \frac{\alpha_c}{\|\nabla \overline{c}\|}\hat{\mathbf{y}}\right) \cdot \nabla \overline{c} - \nabla \cdot \left(\frac{\alpha_b}{(\nabla \overline{c} \cdot \hat{\boldsymbol{\tau}}_b)}(\nabla \overline{c} \cdot \hat{\boldsymbol{\tau}}_b)\hat{\boldsymbol{\tau}}_b\right). \tag{A5}$$

The first term on the right-hand side of (A5) takes the form of an advection operator, in which we can identify the eddy

streamfunction

$$\psi^* = \frac{\alpha_c}{\|\nabla \overline{c}\|} = \frac{\overline{\mathbf{u}'c'} \cdot \nabla b}{\nabla \overline{c} \times \nabla b}. \tag{A6}$$





Note that this definition is ill-defined in the limit $\nabla\bar{c} \times \nabla b \to 0$; in this limit $\hat{\boldsymbol{\tau}}_b$ and $\hat{\boldsymbol{\tau}}_c$ are parallel, the eddy tracer flux is purely advective, and the streamfunction becomes

$$\psi^\star = \frac{\overline{\mathbf{u}'c'} \times \nabla\bar{c}}{\|\nabla\bar{c}\|^2}. \tag{A7}$$

The second term on the right-hand side of (A5) has been written in the form of the divergence of a flux along mean buoyancy surfaces, with the isopycnal gradient operator (see Section 2.1) appearing explicitly as

$$\nabla_{\|} \equiv (\nabla\bar{c} \cdot \hat{\boldsymbol{\tau}}_b)\hat{\boldsymbol{\tau}}_b. \tag{A8}$$

We can then identify the isopycnal diffusivity $\kappa_{\mathrm{iso}}$ as

$$\kappa_{\mathrm{iso}} = -\frac{\alpha_b}{(\nabla\bar{c} \cdot \hat{\boldsymbol{\tau}}_b)} = -(\overline{\mathbf{u}'c'} \cdot \nabla\bar{c})\frac{\|\nabla\bar{b}\|^2}{\|\nabla\bar{b} \times \nabla\bar{c}\|^2} = -\frac{\overline{\mathbf{u}'c'} \cdot \nabla\bar{c}}{\|\nabla\bar{c}\|^2 \cos^2\theta}, \tag{A9}$$

where $\theta$ is the angle between the vectors $\nabla\bar{b}$ and $\nabla\bar{c}$.

While the above derivation is general, for application in MAMEBUS we must make assumptions about the eddy tracer fluxes. Specifically, we assume: (i) that approximately identical eddy streamfunctions $\psi^\star$ advect each different model tracer, and (ii) that the isopycnal diffusivity is positive (*i.e.* that eddy tracer fluxes are always directed down the mean tracer gradients), and (iii) that the isopycnal diffusivity is approximately equal for different model tracers. These assumptions are satisfied in the limit of small-amplitude fluctuations $\mathbf{u}'$ and $c'$ (Plumb, 1979; Plumb and Ferrari, 2005b).

## B    Derivation of Time Variable Adams Bashforth Methods

For a given tracer defined with an associated time tendency equation of the form,

$$\frac{\partial c}{\partial t} = f(t, c(t)). \tag{B1}$$

We integrate Equation (B1) in time from $[t_{n+2}, t_{n+1}]$,

$$\int\limits_{t_{n+1}}^{t_{n+2}} \frac{\partial c}{\partial t} d\tau = \int\limits_{t_{n+1}}^{t_{n+2}} f(\tau, c(\tau)) d\tau. \tag{B2}$$

By the fundamental theorem of calculus,

$$c(t_{n+2}) - c(t_{n+1}) = \int\limits_{t_{n+1}}^{t_{n+2}} f(\tau, c(\tau)) d\tau. \tag{B3}$$

We interpolate the right hand side using a Lagrange polynomial of the form:

$$p(\tau) = \frac{\tau - t_{n+1}}{t_n - t_{n+1}} f(t_n, c(t_n)) + \frac{\tau - t_n}{t_{n+1} - t_n} f(t_{n+1}, c(t_{n+1})). \tag{B4}$$





Then using B4 and substituting it into B3, we have,

$$
\int_{t_{n+1}}^{t_{n+2}} f(\tau, c(\tau)) d\tau = \int_{t_{n+1}}^{t_{n+2}} p(\tau) d\tau
$$

$$
= \int_{t_{n+1}}^{t_{n+2}} \left( \frac{\tau - t_{n+1}}{t_n - t_{n+1}} f(t_n, c(t_n)) + \frac{\tau - t_n}{t_{n+1} - t_n} f(t_{n+1}, c(t_{n+1})) \right) d\tau
$$

$$
= \frac{1}{2} \left[ \frac{(\tau - t_{n+1})^2}{t_n - t_{n+1}} f(t_n, c(t_n)) + \frac{(\tau - t_n)^2}{t_{n+1} - t_n} f(t_{n+1}, c(t_{n+1})) \right]_{t_{n+1}}^{t_{n+2}}.
$$

Defining $\Delta t_{n+1} = t_{n+1} - t_n$, we obtain

$$
\int_{t_{n+1}}^{t_{n+2}} f(\tau, c(\tau)) d\tau = \frac{\Delta t_{n+2}}{2\Delta t_{n+1}} \left( 2f(t_{n+1}, c(t_{n+1})) \Delta t_{n+1} + f(t_{n+1}, c(t_{n+1})) \Delta t_{n+2} - f(t_n, c(t_n)) \Delta t_{n+2} \right). \tag{B5}
$$

Substituting Equation (B3) into Equation (B5) yields the full ABII time stepping scheme, given by Equation (96).

For higher order AB methods, we consider a $s$-th order Lagrange polynomial of the form,

$$
p(\tau) = \sum_{m=0}^{s-1} p_m(\tau) f(t_{n+m}, c(t_{n+m})) \tag{B6}
$$

$$
p_m(\tau) = \prod_{\substack{l=0 \\ l \neq m}}^{s-1} \frac{\tau - t_{n+l}}{t_{n+m} - t_{n+l}} \tag{B7}
$$

where setting $s = 3$ as the number of known points in the interpolating polynomial results in the ABIII method. Then, $s - 1$ is the degree of the polynomial. The general form of higher order AB methods is,

$$
c(t_{n+s}) - c(t_{n+s-1}) = \int_{t+s-1}^{t+s} \sum_{m=0}^{s-1} p_m(\tau) f(t_{n+m}, c(t_{n+m})) d\tau. \tag{B8}
$$

The algebra to solve for the full discrete form of the ABIII method follows the derivation of the ABII method above. The

solution to the integration in Equation (B8) is given by Equation (95).

## C   Comparison of Boundary Layer Parameterizations with Ferrari et al. (2008)

Our representation of eddy advection and isopycnal stirring in the surface mixed layer (SML) and bottom boundary layer (BBL) is adapted from Ferrari et al. (2008), and is described in Section 2.3.2. We now directly compare our SML/BBL scheme against that of Ferrari et al. (2008) to highlight the key differences.





As discussed in Section 2.3.2.1 our SML scheme leads to the same eddy streamfunction as that of Ferrari et al. (2008), given by (29). In contrast, the residual eddy tracer flux in the SML differs as follows

$$(\kappa_{\mathrm{iso}}\nabla_{\parallel}\overline{c})_{\mathrm{FMCD}} = \kappa_{\mathrm{iso}}\left(\overline{c}_x + S_{\mathrm{sml}}\overline{c}_z\right)\hat{\mathbf{x}} + \kappa_{\mathrm{iso}}\left(S_{\mathrm{sml}}\overline{c}_x + S_{\mathrm{sml}}S_{\mathrm{int}}\overline{c}_z\right)\hat{\mathbf{z}}, \tag{C1a}$$

$$\kappa_{\mathrm{iso}}\nabla_{\parallel}\overline{c} = \kappa_{\mathrm{iso}}\left(\overline{c}_x + S_{\mathrm{sml}}\overline{c}_z\right)\hat{\mathbf{x}} + \kappa_{\mathrm{iso}}\left(S_{\mathrm{sml}}\overline{c}_x + S_{\mathrm{sml}}^2\overline{c}_z\right)\hat{\mathbf{z}}, \tag{C1b}$$

where "FMCD" denotes the formulation of Ferrari et al. (2008). Thus our Equation (C3b) differs from Equation (C3a) only by the replacement $S_{\mathrm{int}}$ by $S_{\mathrm{sml}}$ in the vertical eddy residual tracer flux. A drawback of using $S_{\mathrm{int}}$ is that typically the vertical buoyancy gradient is very small in the SML, so the form (C3a) may not be numerically stable. Ferrari et al. (2008) propose a modification of the vertical component of the tracer residual eddy flux to avoid dividing by small vertical buoyancy gradients in the mixed layer,

$$(\kappa_{\mathrm{iso}}\nabla_{\parallel}\overline{c})_{\mathrm{FMCD}}\cdot\hat{\mathbf{z}} = -\kappa_{\mathrm{iso}}\hat{G}(z)\left(-\frac{\overline{b}_x\overline{b}_z}{\overline{b}_z|^2_{z=-H_{\mathrm{sml}}}}\overline{c}_x + \frac{\overline{b}_x^2}{\overline{b}_z|^2_{z=-H_{\mathrm{sml}}}}\overline{c}_z\right)\hat{\mathbf{z}}, \tag{C2}$$

However, this alternative breaks the symmetry of the diffusion tensor, and requires the introduction of an additional vertical structure function, $\hat{G}(z)$. Our formulation, Equation (C3b)m retains the symmetry of the stress tensor and preserves continuity of the vertical flux and its derivative at $z = -H_{\mathrm{sml}}$ with the same structure function $G_{\mathrm{sml}}(z)$ (see Section 2.3.2). It is also simpler to implement, as both the streamfunction (29) and the residual eddy flux (C3b) can be written succinctly in terms of

the effective slope (28).

Another difference between our formulation and that of Ferrari et al. (2008) arises in the eddy stirring of buoyancy in the SML,

$$(\kappa_{\mathrm{iso}}\nabla_{\parallel}\overline{b})_{\mathrm{FMCD}} = \kappa_{\mathrm{iso}}\left(\overline{b}_x + S_{\mathrm{sml}}\overline{b}_z\right)\hat{\mathbf{x}}, \tag{C3a}$$

$$\kappa_{\mathrm{iso}}\nabla_{\parallel}\overline{b} = \kappa_{\mathrm{iso}}\left(\overline{b}_x + S_{\mathrm{sml}}\overline{b}_z\right)\hat{\mathbf{x}} + \kappa_{\mathrm{iso}}\left(S_{\mathrm{sml}}\overline{b}_x + S_{\mathrm{sml}}^2\overline{b}_z\right)\hat{\mathbf{z}}. \tag{C3b}$$

The Ferrari et al. (2008) residual eddy buoyancy flux has no vertical component, whereas ours does. This impacts the rate of available potential energy release in the SML by modifying the total vertical eddy buoyancy flux,

$$\overline{w'b'}_{\mathrm{FMCD}} = \kappa_{\mathrm{gm}}G_{\mathrm{sml}}(z)\frac{\overline{b}_x^2}{\overline{b}_z|_{z=-H_{\mathrm{sml}}}}, \tag{C4a}$$

$$\overline{w'b'} = \kappa_{\mathrm{gm}}G_{\mathrm{sml}}(z)\frac{\overline{b}_x^2}{\overline{b}_z|_{z=-H_{\mathrm{sml}}}} + \kappa_{\mathrm{iso}}G_{\mathrm{sml}}(z)\frac{\overline{b}_x^2}{\overline{b}_z|_{z=-H_{\mathrm{sml}}}}\left(1 - G_{\mathrm{sml}}(z)\frac{\overline{b}_z}{\overline{b}_z|_{z=-H_{\mathrm{sml}}}}\right), \tag{C4b}$$

The key difference here is that our version (C5b) typically releases more potential energy, and is not strictly positive definite;

if $\overline{b}_z \gg \overline{b}_z|_{z=-H_{\mathrm{sml}}}$ then in principle $\overline{w'b'}$ may be negative. This corresponds to creation of potential energy, whereas previous studies suggest that potential energy should be consistently released in the SML (Colas et al., 2013). However, by construction, the vertical derivative of this term is zero at $Z = -H_{\mathrm{sml}}$, and in any practical case $\overline{b}_z$ will be smaller than $\overline{b}_z|_{z=-H_{\mathrm{sml}}}$ throughout the boundary layer. This suggests that if the vertical eddy lengthscale $\lambda$ is positive $\overline{b}_z z < 0$ then our scheme releases potential energy everywhere. Note also that the GM component of the vertical eddy buoyancy flux always releases potential energy.





Finally, we compare the horizontal component of the eddy buoyancy flux in the SML,

$$\overline{u'b'}_{\text{FMCD}} = -\kappa_{\text{gm}}\overline{b}_x, \tag{C5a}$$

$$\overline{u'b'} = -\kappa_{\text{iso}}\overline{b}_x + (\kappa_{\text{gm}} - \kappa_{\text{iso}})S_{\text{sml}}\overline{b}_z. \tag{C5b}$$

Whereas the Ferrari et al. (2008) scheme preserves strict lateral downgradient diffusion, this is only true in our scheme if

$\kappa_{\text{gm}} = \kappa_{\text{iso}}$.

Further to this comparison with the formulation of Ferrari et al. (2008), we note that the fluxes discussed above differ substantially in the BBL over sloping topography. For example, the vertical buoyancy flux becomes

$$\overline{w'b'} = \kappa_{\text{gm}}G_{\text{bbl}}(z)\frac{\overline{b}_x^2}{\overline{b}_z\big|_{z=\overline{\eta}_b+H_{\text{bbl}}}} - \kappa_{\text{iso}}\widetilde{S}_e\overline{b}_z(\widetilde{S}_e - S_{\text{int}}) \tag{C6}$$

Thus in general the eddy buoyancy flux will act to create potential energy ($\overline{w'b'} < 0$) unless the isopycnal slope $S_{\text{int}}$ is of the

same sign as the bottom slope and larger in magnitude. In order to avoid this, it would be necessary to set $\kappa_{\text{iso}} = 0$ throughout the BBL. This is a separate consideration from the orientation of the residual flux vector, which must certainly lie parallel to the topography if the diffusivity is nonzero.

*Author contributions.* Andrew Stewart conceived and coordinated the development of MAMEBUSv1.0, and advised Jordyn in further model development. Jordyn Moscoso developed and implemented the ecosystem models, updated the calculation of the momentum equation, and

fine-tuned the pressure gradient calculations. Daniele Bianchi coodinated the development of the ecosystem model. Jim McWilliams coordinated the development of the eddy parameterizations in MAMEBUS and advised Jordyn in further model development. Jordyn Moscoso prepared the manuscript with contributions from all co-authors.

*Competing interests.* The authors declare no competing interests

*Acknowledgements.* This material is based in part upon work supported by the National Science Foundation with grant numbers OCE-

1538702, OCE-1751386, and OCE-1635632, and by the National Aeronautics and Space Administration ROSES Physical Oceanography program under grant number 80NSSC19K1192. D.B. gratefully acknowledges funding from the Alfred P. Sloan Foundation. This work used the Extreme Science and Engineering Discovery Environment (XSEDE, Towns et al. 2014), which is supported by National Science Foundation grant number ACI-1548562.





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
