# Peer review of "A Meridionally Averaged Model of Eastern Boundary Upwelling Systems (MAMEBUSv1.0)"

_Geoscientific Model Development, 2020_

## Referee Comment (RC1) · Anonymous Referee #1 · 22 Oct 2020

Moscoso et al present a newly written ocean circulation biogeochemical model, NAME-BUS, written in C, with Matlab scripts added to set up and visualize the model and its solutions. The model is configured to represent a two dimensional eastern boundary upwelling system. With the idealized set up the computational efficiency is high, which in turn allows to relatively quickly carry out many simulations. An evaluation of the model (set up as the California Current System) against observations indicates that the model reproduces major features of the upwelling system.

The draft is written very well, carefully and clearly. The high model's computational efficiency (it can be run on a laptop/desktop) allows to test physical or biogeochemical parameterizations/model formulations, to run sensitivity studies, or to carry out parameter optimizations. I.e., NAMEBUS promises to be a useful tool that nicely complements

realistic three dimensional set ups for EBUS. I recommend the draft for publication.

**Detailed comments:**

- wind stress forcing: I expect the wind stress maximum to be located somewhat offshore, as indicated in your own Fig 1, see also, e.g,. SCOW data in http://www.clivar.org/sites/default/files/EBUS-Prospectus.pdf (Fig 2) or Fennel et al, 2012, https://www.sciencedirect.com/science/article/pii/S0278434312001525 (Fig 3), or also in Castelao and Luo, 2018, that you refer to. This contrasts Fig 5 in your paper which shows maximum wind stress at the coast if I see correctly. Would it be valuable to introduce a parameter that allows to set the offshore distance of the wind stress maximum, similar to the parameter l in Fig 7 of Fennel et al, 2012?

- given that you submitted to GMD and that the computational efficiency of the model is a major point, I suggest to provide in section 4 "Implementation Details" some additional information. What software/Matlab toolboxes/etc are required to run the model, what minimum hardware requirement one needs, if it is platform independent (operating system, laptop/desktop/linux cluster... I presume it is not parallelized?). I.e., if I understand correctly, NAMEBUS' advantage is that you can simply run it on a laptop (?) and that it is easily transferable. I think it is worth to stress this value. The model could even be used for educational purposes, couldn't it? Maybe also provide some benchmark number for computation time (to get a sense for, e.g,. how long it will take me if I want to run a parameter optimization).

- P39 L16/17 "This underestimate may be caused by the over-simplification of the ecosystem structure in the NPZD model": The underestimate could also originate

in differences due to the circulation, couldn't it (as the pattern of the Chl bias in Fig 5 bottom panel is matched by a bias in temperature, Fig 5 top panel)?

- P39 L27 "5.4 Model Verification (Resolution Tests)": This title is not entirely clear to me. Also the subsection consists only of a single paragraph and the associated Figure is hardly discussed. I suggest to either delete or add more detail to this subsection.

**Grammar:**

- P40 L3 "The model is based on solution of a general evolution equations for": Needs rephrasing

- P40 L8 "we performeda preliminary": Typo

---

## Referee Comment (RC2) · Anonymous Referee #2 · 27 Oct 2020

In this manuscript, Moscoso et al. present a new numerical approach for modeling the physics and planktonic ecosystems of eastern boundary upwelling systems. Their approach is model variations in two dimensions (depth and offshore distance) while parameterizing some of the important effects of alongshore variability. The effects of cross shore mass fluxes driven by eddies are parameterized using long-standing methods (e.g. Gent and McWilliams). A method for incorporating the effects of an alongshore pressure gradient are also described.

The model is compared with a hydrographic section from the CalCOFI program, demonstrating agreement with general patterns offshore of the continental shelf (upwelling isotherms, subsurface chlorophyll maximum and reasonable values for temperature and nitrate).

[Figure]

This is an interesting and new approach, and it should be useful to the coastal oceanography community. The 2D approach could be used for idealized experiments or as a test bed for additional ecosystem models beyond the NPZD model in this initial version. I can envision adding oxygen or other nutrients. It fills a niche that is distinct from 3D models like ROMS, which can be run in a 2D mode but do not easily allow for an alongshore pressure gradient. The study is interesting, well-executed and clearly presented. However, I do have some points that the authors may wish to consider addressing when discussing the limitations of this simplified approach.

Detailed comments:

p. 6, lines 12-4 – The momentum equations neglect the nonlinear terms. However, in upwelling regions characterized by strong stratification and steep bottom slopes, the momentum advection term can be an important part of the alongshore momentum balance (Lentz and Chapman 2004). This impacts the source depth for upwelling, which in turn affects nitrate concentrations on the shelf (Jacox and Edwards, 2011). The limitations of this assumption should be discussed.

p. 25, last line – numerical methods for calculating the zonal pressure gradient force in sigma coordinates are described in detail. Have associated errors been evaluated for the reference solution? This is often done by running the model with initial stratification but no forcing.

In Section 3.5.2, the methods for imposing an alongshore pressure gradient are discussed in detail, and this is one of the motivations for developing this model. Has the implementation of the alongshore pressure gradient been tested? This might improve the density structure, as well as the meridional velocity. The CalCOFI observations indicate downward tilting isotherms near the continental slope at depths > 150m, a signature of a poleward undercurrent. A poleward pressure gradient can be associated with an onshore geostrophic flow near the surface.

The model output shows a well-mixed region over the shelf, which does not seem

realistic (although the bathymetry differs between the model and observations). This feature should be discussed. It may have to do with the specification of 40m deep surface and bottom mixed layers, which overlap on the shelf. Is a surface buoyancy flux included in the test run?

---

## Author Response (AR1)

We thank both reviewers for their constructive and thoughtful comments and for taking the time to read our manuscript. Your comments have improved the clarity and discussion in our paper.

The reviewers comments are listed point-by-point in red. Our responses are listed in blue. Additional changes to the manuscript are listed in black with associated page and line numbers.

**Response to Referee 1**

**Detailed Comments:**

**Comment**: wind stress forcing: I expect the wind stress maximum to be located some- what offshore, as indicated in your own Fig 1, see also, e.g,. SCOW data in http://www.clivar.org/sites/default/files/EBUS-Prospectus.pdf (Fig 2) or Fennel et al, 2012, https://www.sciencedirect.com/science/article/pii/S0278434312001525 (Fig 3), or also in Castelao and Luo, 2018, that you refer to. This contrasts Fig 5 in your paper which shows maximum wind stress at the coast if I see correctly. Would it be valuable to introduce a parameter that allows to set the offshore dis- tance of the wind stress maximum, similar to the parameter l in Fig 7 of Fennel et al, 2012?

Thank you for your suggestion, we plan to make this aspect more clear for other readers. The wind-stress forcing that is presented in equation (125) can be updated and defined by the user in any way that they see fit, the shape of the profile and wind-stress maximum can be given to the model in any form, e.g. based on observations. We have made this more clear by adding the following to the description on choice of wind-stress forcing:

**(p. 35 L.5-7)** "We tune the offshore maximum to approximate values reported by Castealo and Luo (2018). While this is the example of wind-stress forcing we choose to use to validate our model, any form of wind-stress forcing can be defined by the user."

**Comment:** given that you submitted to GMD and that the computational efficiency of the model is a major point, I suggest to provide in section 4 "Implementation Details" some addi- tional information. What software/Matlab toolboxes/etc are required to run the model, what minimum hardware requirement one needs, if it is platform independent (operating system, lap- top/desktop/linux cluster... I presume it is not parallelized?). I.e., if I understand correctly, NAMEBUS' advantage is that you can simply run it on a laptop (?) and that it is easily trans- ferable. I think it is worth to stress this value. The model could even be used for educational purposes, couldn't it? Maybe also provide some benchmark number for computation time (to get a sense for, e.g,. how long it will take me if I want to run a parameter optimization).

This is a great suggestion. We have added more detail in the paper as suggested with the following details:

- Matlab packages required to run the model. (Section 4)

- Software required to run the model locally on a laptop or desktop (Section 4)

- Computational times given increasing grid resolution on the cluster and a 2015 Mac Laptop. (Section 5.4, Table 6)

**Comment:** P39 L16/17 "This underestimate may be caused by the over-simplification of the ecosystem structure in the NPZD model": The underestimate could also originate in differences due to the circulation, couldn't it (as the pattern of the Chl bias in Fig 5 bottom panel is matched by a bias in temperature, Fig 5 top panel)?

Yes, that is a good point, we have added the following comment to address this:

**(p. 39 L. 1-4)** "This underestimate may be caused by the over-simplification of the ecosystem structure in the NPZD model, which only includes a single phytoplankton group, while multiple groups are likely required for a more correct representation of enhanced coastal phytoplankton biomass (Van Oostende et. at., 2018). Furthermore, aspects of these differences could be caused by the idealized nature of the 2-D circulation simulated by the physical model."

**Comment:** P39 L27 "5.4 Model Verification (Resolution Tests)": This title is not entirely clear to me. Also the subsection consists only of a single paragraph and the associated Figure is hardly discussed. I suggest to either delete or add more detail to this subsection.

Thank you for your feedback on this. We have updated the title of Section 5.4 to be Resolution Parameter Sweep, and have expanded this section to include more description listed below:

**(p.40 - 2-14)** "In this section, we describe the changes in solutions due to model resolution. We chose four different resolutions, and explored the results. Figure 9 shows the solutions of MAMEBUS after 30 model years. Each panel shows the model state in the euphotic zone, averaged over the final 10 years of integration . All resolutions have the same setup and forcing as described in Sections 4 and 5. The top row shows the potential temperature ($\theta$), the middle row shows the nitrate concentration, and the bottom row shows the phytoplankton concentration. The model grid resolution increases from left to right, with the coarsest simulation run on a grid of 32 points horizontally and vertically, and the highest-resolution simulation run on a grid of 128 points horizontally and vertically.

Increasing the resolution leads to an overall shoaling of nutrients toward the surface. The largest overall change in near-slope nutrient concentration occurs when the resolution doubles from 32 to 64 horizontal points and vertical levels. Increasing the resolution beyond a 64x64 grid does not substantially change the horizontal distribution of phytoplankton . As referenced in Table 6, doubling the resolution increases the model run time by a multiple of approximately 20. Thus while the model can practically be run at higher resolution, our tests show that intermediate resolution (64 horizontal and vertical levels) is sufficient to produce a favorable comparison with in-situ data, without substantially increasing the computation time."

**Grammar:**

P40 L3 "The model is based on solution of a general evolution equations for": Needs rephrasing

We have changed this to:

**(P. 20 L 3-5)** "The solutions are determined by a general evolution equation for materially conserved tracers (Section 2) and the fluid momentum equations and the fluid momentum equations under the time-dependent turbulent thermal wind (T3W) approximation (Dauhajre and McWilliams, 2018)."

P40 L8 "we performeda preliminary": Typo

Fixed. Thank you.

**Response to Referee 2**

**Detailed Comments:**

**Comment:** p. 6, lines 12-4 – The momentum equations neglect the nonlinear terms. However, in upwelling regions characterized by strong stratification and steep bottom slopes, the momentum advection term can be an important part of the alongshore momentum balance (Lentz and Chapman 2004). This impacts the source depth for upwelling, which in turn affects nitrate concentrations on the shelf (Jacox and Edwards, 2011). The limitations of this assumption should be discussed.

This is indeed a limitation of this model. We neglected momentum advection under the assumption of small-Rossby number flow, with the practical outcome of simplifying the formulation of this model. To address this assumption we added the following discussion:

**(p.6 L.14-17)** "This assumption may indeed have some limitations in upwelling regions with steep topography and strong stratification. Lentz and Chapman (2004) show that in the cross-shelf momentum flux divergence balances the wind-stress and supports an on shore return flow, which can impact nitrate concentrations on the shelf (Jacox and Edwards, 2011)."

**Comment:** 25, last line – numerical methods for calculating the zonal pressure gradient force in sigma coordinates are described in detail. Have associated errors been evaluated for the reference solution? This is often done by running the model with initial stratification but no forcing.

Indeed, we have evaluated errors in the pressure/buoyancy gradients with the cubic scheme. This has been extensively tested in Shchepetkin and McWilliams (2003), and within ROMS, and because of this we do not include this test case in the manuscript – however we have added the following text to the manuscript:

**(p.25 L.26/27- p.26 L.1-5)** "Pressure gradient calculations in sigma coordinates have been long known to produce discretization errors from the misalignment of geopotential and sigma coordinate surfaces and rely on large cancellations in the vertical gradient near steep slopes (Arakawa and Suarez, 1983; Haney, 1991; Mellor et al., 1994, 1998). We follow Shchepetkin and McWilliams (2003) to calculate the horizontal pressure gradient force and reduce the errors in horizontal gradient calculations, which otherwise produce large spurious along-slope currents in MAMEBUS (not shown) This algorithm has been extensively tested via its implementation in ROMS (Shchepetkin and McWilliams, 2003; 2005) so we omit our own tests of the pressure gradient calculation scheme in this study."

For the reviewer's reference, here is an example figures showing the cubic gradient scheme amd associated errors:

[Figure]

Figure 1: A figure showing the buoyancy gradient from the initial buoyany profile defined in Equation 127.

The left panel shows the analytical buoyancy gradient from the initial buoyancy profile. The center panel shows the linear buoyancy gradient calculated using Equation 68a. The rightmost panel shows the cubic buoyancy gradient calculated following the algorithm of Shchepetkin and McWilliams (2003) and described in Section 3.5. The following figure shows the error in both the linear and cubic gradient schemes.

[Figure]

Figure 2: A figure showing the error associated with the linear and cubic buoyancy gradient schemes over steep topography

The linear scheme overestimates the buoyancy gradient near the slope due to large grid-scale cancellations in the horizontal buoyancy gradient. This is consistent with the findings in Arakawa and Suarez (1983), Haney (1991), Mellor et al. (1994, 1998), and Shchepetkin and McWilliams (2003). The cubic scheme produces substantially smaller errors in regions of steep slopes due to the higher order interpolation scheme.

**Comment:** In Section 3.5.2, the methods for imposing an alongshore pressure gradient are discussed in detail, and this is one of the motivations for developing this model. Has the implementation of the alongshore pressure gradient been tested? This might improve the density structure, as well as the meridional velocity. The CalCOFI observations indicate downward tilting isotherms near the continental slope at depths > 150m, a signature of a poleward undercurrent. A poleward pressure gradient can be associated with an onshore geostrophic flow near the surface.

We have tested the influence of along-slope pressure gradients, but omitted them from our reference solution in the interest of simplicity. We address this issue in discussion of our future work (Section 6).

On p.40 L 6-9 we state:
"An aspect of MAMEBUS that requires further investigation is the effect of meridional pressure gradients, which we neglected in our reference solutions in Section 5. In reality, the presence of along-shore pressure gradients may support interior across- shore transport away from the surface and bottom boundary layers, with the potential to reshape the coastal ecosystem."

However we have added the following in the discussion of the meridional pressure gradients in section 3.5.2 to address this more clearly:

**(p.29 L.18-21)** "Though MAMEBUS allows meridional pressure gradients to be imposed, we have excluded them from our reference solutions in the interest of simplicity. However, previous studies have highlighted the importance of meridional pressure gradients in supporting interior cross-slope transport, and in driving poleward undercurrents (Connelly et. al., 2014). We plan to address the effects of meridional pressure gradients on EBUS ecosystem dynamics in future scientific studies using MAMEBUS."

For the reviewer's reference, we have done some preliminary testing using barotropic pressure gradients, shown below. These runs show the mean overturning stream function given a change of sea surface height of 0.1 cm (top row) and 1 cm (bottom row) over a 1000km meridional distance. The following figures are illustrations of the mean circulation given a meridional barotropic pressure gradient as a result of some realistic variations in SSH – approximations are inferred from Renault, Lionel, et al. (2020) - bioRxiv (https://www.biorxiv.org/content/10.1101/2020.02.10.942730v1.full). These examples should be interpreted as illustrations of model behavior, and not as a realistic representation of the mean zonal circulation.

[Figure]

Figure 3: The topmost figure shows the mean circulation from Figure 8 (p.40). The left column on the four panel figure shows the mean zonal stream function given a northward/poleward pressure gradient, and the right column shows the mean overturning circulation given a southward/equatorward pressure gradient. The figures in the top row correspond to a change in sea surface height of about 0.1 cm over 1000km, and the bottom row corresponds to a change in SSH of about 1cm over 1000km. The regions with negative circulation (blue) correspond to counterclockwise circulation, the regions with positive circulation (red) correspond to clockwise circulation.

This illustration is not a realistic example of the circulation because the along-shore pressure gradient is likely to be concentrated along the shelf and the slope and not have much effect offshore (Connelly et. al. 2014). However, with the addition of the barotropic pressure gradient we indeed get internal onshore (and offshore) flow given a poleward (equatorward) barotropic pressure gradient. On long timescales the poleward pressure gradient is balanced by the coriolis force and supports a zonal onshore flow at depth. The input of wind-stress is balanced by the pressure gradient and the onshore transport is no longer isolated to the bottom boundary layer. The equatorward pressure gradient shows an interior offshore transport and an intensification of upwelling isolated to the bottom boundary layer.

For this model description paper, we opt not to show these experiments because they are not tuned to represent realistic or idealized interior circulation. This functionality allows us to refine aspects to represent different EBUs with important aspects (such as meridional pressure gradients) present in future studies.

**Comment:** The model output shows a well-mixed region over the shelf, which does not seem realistic (although the bathymetry differs between the model and observations). This feature should be discussed. It may have to do with the specification of 40m deep surface and bottom mixed layers, which overlap on the shelf.

That is a good point. The model topography does differ from the observation specifically in the inner shelf which is too wide and deep compared to observation. Because of our formulation any shelf shallower than 80m would be very well mixed. Indeed thinner mixed layer specifications would reduce the amount of mixing on the shelf. We address this on p.40 L5-6: "Furthermore, we plan to expand upon the physical framework in this paper by expanding eddy parameterizations to include the effect of submesoscale eddies on the shelf, where the mesoscale eddy activity is inhibited." However, we had added the following in the discussion of the solutions to address this more clearly:

**(p.38 L10-18)** "Furthermore, we prescribed a continental shelf that is deeper than in nature in order to reduce the model's computation time. Further shallowing the continental shelf is possible, but the CFL constraint imposed by the finer vertical resolution on the shelf extends the computation time. While the continental slope is tuned to have a similar slope as observations in Central California near the shelf break, the mixed layers in this model run are set to a constant depth zonally and overlap on the shelf. This choice has been made for simplicity, and could be refined via zonally-varying mixed layer depths to improve agreement with specific EBUSs. The well mixed area on the shelf is an analogue to the inner shelf, albeit somewhat deeper than those found in nature (Lentz and Fewings, 2012). In our model comparison, we neglect the inner shelf region in the model and compare the solutions and starting approximately 50km from the coast."

**Comment:** Is a surface buoyancy flux included in the test run?

Yes! There is a surface buoyancy flux in the form of a surface restoring to the initial profile at the surface grid box. The discussion of restoring is given on p. 36 lines 15-20. To make this more clear, we added a comment at line 20.

[revised manuscript text omitted]

$$\kappa_{\text{dia}}(x, z, t) = \kappa_{\text{sml}}(x, z) + \kappa_{\text{bbl}}(x, z) + \kappa_{\text{conv}}(x, z, t) + \kappa_{\text{bg}}(x, z). \tag{15}$$

The terms on the right-hand side of (15) are discussed in turn in the following paragraphs.

The diapycnal diffusivity in the surface mixed layer, $\kappa_{\text{sml}}$, is prescribed to have the same structure as that used in the K-profile parameterization (KPP) of Large et al. (1994). However, for simplicity, the mixed layer depth $H_{\text{sml}}(x)$ and maximum magnitude $\kappa_{\text{sml}}(x)$ are prescribed functions, rather than depending on the local surface forcing. The vertical profile of $\kappa_{\text{dia}}$ in the surface mixed layer, *i.e.* $-H_{\text{sml}} < z < 0$, is given by

$$\kappa_{\text{sml}}(x, z) = \kappa_{\text{sml}}^0 G_{\text{KPP}}(\sigma_{\text{sml}}), \tag{16}$$

where the dimensionless surface mixed layer vertical coordinate $\sigma_{\text{sml}} = -z/H_{\text{sml}}$ is defined such that $0 \leq \sigma_{\text{sml}} \leq 1$ within the mixed layer. The structure function $G_{\text{KPP}}(\sigma_{\text{sml}})$ is given by,

$$G_{\text{KPP}}(\sigma) = \begin{cases} \dfrac{27}{4}\sigma_{\text{sml}}(1 - \sigma_{\text{sml}})^2, & 0 \leq \sigma_{\text{sml}} \leq 1, \\ 0, & \sigma_{\text{
[revised manuscript text omitted]